# Cost Efficient Fairness Audit Under Partial Feedback

**Nirjhar Das**
Indian Institute of Science
nirjhardas@iisc.ac.in

**Mohit Sharma**
IIIT Delhi
mohits@iiitd.ac.in

**Praharsh Nanavati**
Indian Institute of Science
praharsh.nanavati@gmail.com

**Kirankumar Shiragur**
Microsoft Research India
kshiragur@microsoft.com

**Amit Deshpande**
Microsoft Research India
amitdesh@microsoft.com

## Abstract

We study the problem of auditing the fairness of a given classifier under partial feedback, where true labels are available only for positively classified individuals, (e.g., loan repayment outcomes are observed only for approved applicants). We introduce a novel cost model for acquiring additional labeled data, designed to more accurately reflect real-world costs such as credit assessment, loan processing, and potential defaults. Our goal is to find optimal fairness audit algorithms that are more cost-effective than random exploration and natural baselines.

In our work, we consider two audit settings: a black-box model with no assumptions on the data distribution, and a mixture model, where features and true labels follow a mixture of exponential family distributions. In the black-box setting, we propose a near-optimal auditing algorithm under mild assumptions and show that a natural baseline can be strictly suboptimal. In the mixture model setting, we design a novel algorithm that achieves significantly lower audit cost than the black-box case. Our approach leverages prior work on learning from truncated samples and maximum-a-posteriori oracles, and extends known results on spherical Gaussian mixtures to handle exponential family mixtures, which may be of independent interest. Moreover, our algorithms apply to popular fairness metrics including demographic parity, equal opportunity, and equalized odds. Empirically, we demonstrate strong performance of our algorithms on real-world fair classification datasets like Adult Income and Law School, consistently outperforming natural baselines by around 50% in terms of audit cost.

## 1   Introduction

Machine learning models can amplify and exacerbate biases present in their training data. To address this, fairness audits and interventions have emerged as key topics in the fair machine learning literature [MMS+21, BHN23, YZC+24]. Consequently, a wide range of fairness metrics [DHP+12, HPS16, GVF20], fairness interventions [BDH+19, BDE+20], and audit frameworks have been proposed and adopted [SKH+18, HWVDI+19].

This work addresses the problem of auditing machine learning models for fairness, an issue of growing significance in decision-making systems [BG18, LB23, MSKF24]. Previous work on fairness audit covers a range of individual and group fairness metrics [XYS20, KNRW18] and construction of statistical tests for auditing [TBKN21, SMBN21]. In the simplest fairness audit setting, a decision-maker employs a classifier $f$ that takes as input a feature vector $x$ and a sensitive attribute or group label $a$ and makes a binary prediction $f(x, a)$. An auditor aims to estimate the fairness of $f$ using a standard fairness metric like Equalized Odds, Equal Opportunity with respect to sensitive attributes.

39th Conference on Neural Information Processing Systems (NeurIPS 2025) Workshop: Reliable ML.

In many real-world settings such as a bank disbursing loans using a classifier $f$, the class label $y$ indicating the loan repayment status is available only for applicants who are offered loans (*i.e.*, when $f(x, a) = 1$). Thus, the historical data is biased or incomplete, whereas any new data exploration can be costly (e.g., random disbursal of loans has higher defaults). Previous work under this incomplete data model can be broadly categorized into two settings. First, learning fair classifiers from incomplete labeled data (along with unlabeled data) without actively acquiring the full set of labels [LKL$^+$17, KZ18, KRS$^+$20, Wei21]. Second, online fair classification with *partial feedback*, where labels are revealed only for positively classified data points [BLR$^+$19, KMC24]. Complementary to these works, we study the fairness audit problem in the partial feedback setting under a novel cost model that incorporates both access to historically biased or incomplete data and the cost of acquiring additional labels to complete it.

In the partial feedback setting, due to the unavailability of labels for negatively classified individuals ($f(x, a) = 0$), any fairness audit algorithm must acquire these labels to complete the data and accurately estimate outcome-based fairness metrics such as equal opportunity and equalized odds on the true data distribution. Notably, in scenarios such as bank loan applications, granting a loan to a previously negatively classified individual may incur a cost if the individual fails to repay. This cost is not modeled in prior fairness audit work, which primarily focuses on hypothesis testing for a given unfairness tolerance [KNRW18, XYS20].

**Our Contributions**: We study the problem of fairness audit under partial feedback with a novel cost model. Our cost model captures a central concern of real-world scenarios, where there is a price to pay when the auditor decides to approve an applicant with negative outcome (e.g., loan default). For our cost model, we provide a complete characterization of the sample and cost complexity required for auditing fairness in the non-parametric setting by establishing a lower bound and proposing an audit algorithm that matches this bound up to logarithmic factors.

To further investigate auditing with partial feedback, we study a structured setting where the group-wise distribution of features and labels follows a mixture of exponential families, a common analytical assumption in the fairness literature [DWY$^+$20, MWL22]. In this setting, we provide a novel perspective by viewing the classifier $f$ as inducing a truncation on the input distribution by positively classifying only a subset of samples. We leverage tools from learning with truncated samples [DGTZ18, LWZ23] to address the fairness audit problem under partial feedback. Furthermore, we generalize prior work on maximum a-posteriori (MAP) oracles for mixtures of spherical Gaussians [Mah11] to mixtures of exponential families, which may be of independent interest.

## 1.1 Related work

**Learning with Partial Feedback**: The problem of learning from selection-biased or partial feedback data has been rigorously studied in various contexts in past works. [LKL$^+$17] study the problem of evaluation with selection bias and propose using the heterogeneity of available human decision makers to account for unobserved data. [KZ18] study the residual unfairness of models trained on selection-biased data and propose to correct for sample selection bias by reweighting using benchmark data. [DADC18] leverages the observation that human experts can exhibit consistent prediction in certain regions and uses this to design a data augmentation scheme. [KRS$^+$20] propose using exploration-based stochastic decision rules over predictive models trained on selection-biased data, to achieve fairness and utility with sample selection bias. [Wei21] study utility maximization under selection bias in an MDP setup. Finally, [YLN22] proposes an active debiasing model that adjusts a classifier initially learned on a partial feedback dataset, and where the predictor evolves over time with bounded exploration.

[BLR$^+$19, BJW20, Bec24] study fair online classification for group and individual fairness metrics under the partial feedback setting. Similarly, [KMC24] studies the problem of data collection and fair classification across all subgroups in a partial feedback setting. Complementary to these learning problems, our objective is to construct a cost-efficient algorithm to audit a decision maker for fairness using a framework that combines past, selection-biased data, and online exploration. To the best of our knowledge, our work is the first to define and minimize cost and sample complexity while testing for equalized odds under partial feedback. We also note our work is different from the work on fairness with missing data [ZL21, MQH$^+$25] where the objective is to work with missing covariates but full outcome information, unlike partial feedback.

**Fairness Audit**: Fairness auditing is a well-studied research topic, with practical toolkits for auditing [BDH$^+$19, SKH$^+$18], quantifying sample sizes [SC23, SXK$^+$23], and theoretical frameworks. [CCGWR23] propose a sequential hypothesis testing framework with the ability to ingest data continuously and handle drifts in the data. [SMBN21, TBKN21, GMD$^+$25] leverage optimal transport techniques to construct statistical tests and then enforce exact and approximate fairness. [YZ22] study the auditing problem in a realizable setting via active queries. [BSO$^+$24] study fairness audit with missing confounders for treatment rate estimation. Other seminal works in fairness auditing literature handle potentially infinite and a rich collection of subgroups [KNRW18, MOW$^+$19, CC24]. To the best of our knowledge, we are the first to study fairness audit under partial feedback.

**General Notations.** We denote random variables by upper case letters, e.g., $X, Y$. We use $\mathbb{1}\{\mathcal{F}\}$ to denote the indicator of the event $\mathcal{F}$. $\mathbb{P}[\mathcal{F}]$ and $\mathbb{E}[X]$ respectively denote the probability of the event $\mathcal{E}$ and the expected value of the random variable $X$. With $v \in \mathbb{R}^d$ and $c > 0$, we denote by $\mathcal{B}(v, c)$ the Euclidean ball of radius $c$ around $v$. For two reals $a$ and $b$, we denote $a \vee b = \max\{a, b\}$ and $a \wedge b = \min\{a, b\}$. We use $\widetilde{O}(\cdot)$ and $\widetilde{\Omega}(\cdot)$ to denote the big-O notations and big-Omega notation respectively while hiding log-factors and $\log(1/\delta)$.

## 2   Problem Formulation

In the fairness audit problem, we are interested in testing the fairness of a given *classifier* $f : \mathcal{X} \times \mathcal{A} \to \{0, 1\}$. Here $\mathcal{X}$ denotes the set of *features* and $\mathcal{A}$ denotes the group labels of individuals. Every individual is characterized by a random tuple $(X, A, Y)$, $X \in \mathcal{X}$, $A \in \mathcal{A}$ and $Y \in \{0, 1\}$, where $Y$ is the *true label* of the individual. If $f(X, A) = 1$ for an individual, the individual is classified as positive; otherwise, they are classified as negative. The distribution $\mathcal{D}$ of $(X, A, Y)$ is unknown to the auditor. We adopt *equalized odds* as our primary fairness notion for the remainder of the paper, although our techniques can also be applied to other fairness metrics such as demographic parity [DHP$^+$12] and equal opportunity [HPS16].

Throughout the paper, we use the shorthand notation $f := f(X, A)$ whenever there is no ambiguity regarding $X$ and $A$. We define $\mathbb{P}[f \mid Y, A] := \mathbb{P}_X[f(X, A) \mid Y, A]$.. The *equalized odds difference* (EOD) is then defined as [HPS16]:

$$\Delta := \max_{y \in \{0,1\}} \max_{a, a' \in \mathcal{A}} \left| \mathbb{P}[f = 1 | Y = y, A = a] - \mathbb{P}[f = 1 | Y = y, A = a'] \right|$$

Under this notion, a classifier $f$ is said to be fair if $\Delta = 0$ and unfair if $\Delta > \varepsilon$ for some $\varepsilon > 0$. We pose the fairness audit of $f$ as a hypothesis testing problem that we now describe.

**Definition 1** (($\varepsilon, \delta$)-Fairness Audit)**.** *Given $\varepsilon, \delta \in (0, 1)$, an algorithm/auditor is said to succeed in the ($\varepsilon, \delta$)-fairness audit if with probability at least $1 - \delta$ it can identify correctly which of the following hypotheses is true: (i)* FAIR : $\Delta = 0$*, OR (ii)* UNFAIR : $\Delta > \varepsilon$.

The parameter $\varepsilon$ can be seen as a fairness threshold which the classifier must comply with and whose value can be chosen according to the requirement of the problem.

**Partial Feedback:** We now describe the ($\varepsilon, \delta$)-fairness audit under the partial feedback (PF) model. In the vanilla PF model, the true label is revealed for positively classified individuals only, e.g., the loan default ($Y \in \{0, 1\}$) can only be observed after the loan is given ($f = 1$) by the bank. However, we allow the auditor to request true labels by overriding the negative classification made by the decision-maker. Hence the auditor can explore additional data, which is necessary for the following reason. In our model, an auditor can access $(X, A, Y)$ of individuals who were classified positively in the past by $f$, but only $(X, A)$ of individuals who were classified negatively. From this database, the probability $\mathbb{P}[f = 1, Y = y, A = a]$, can be estimated, but not $\mathbb{P}[f = 0, Y = y, A = a]$, $y \in \{0, 1\}$ and $a \in \mathcal{A}$, since $Y$ is unknown when $f = 0$. We therefore let the auditor access online samples from $\mathcal{D}$ and request $X$ and $Y$ of negatively classified individuals. With this combination of past and online data, the auditor must succeed in the ($\varepsilon, \delta$)-fairness audit.

**Cost model:** We now introduce a novel but natural cost model motivated by practical examples of the partial feedback setting like bank loan applications [KHD22], criminal recidivism [ALMK22], and college admissions [CDGN$^+$23]. The auditor is charged a fixed cost $c_{feat} \geq 0$ every time they request only the feature of an individual for whom $f = 0$. If the auditor requests only the true label or both the feature and the true label of such an individual, they are charged $c_{feat} + c_{lab} \cdot \mathbb{1}\{Y = 0\}$, $c_{lab} \geq 0$. We illustrate the motivation of this cost definition through the example of bank loan

application. Firstly, there may be a cost involved in collecting the data of an individual, which is captured by $c_{feat}$. Secondly, to obtain the true label, the auditor has to force the bank (since $f = 0$) to give loan to the individual. If the individual fails to return the loan ($Y = 0$), only then the bank suffers a monetary loss. To capture this asymmetry, we define the cost formally below.

**Definition 2** (Audit Cost). Let the auditor conclude the $(\varepsilon, \delta)$-fairness audit after running their algorithm on $N$ iid individuals sampled from $\mathcal{D}$ and let $g_i$ and $h_i$ respectively denote whether the auditor observes the feature only and the true label of the $i$-th individual. Then, the audit cost is:

$$\texttt{cost} := \sum_{i=1}^{N} \mathbb{E}\big[c_{feat} \cdot \mathbb{1}\{g_i = 1 \text{ or } h_i = 1, f = 0\} + c_{lab} \cdot \mathbb{1}\{h_i = 1, Y = 0, f = 0\}\big].$$

Typically, $c_{feat} \ll c_{lab}$, e.g. in the bank loan application, the cost of loan default is much higher than the loan processing fee, credit risk assessment etc. For the partial feedback setting, the audit cost is a more effective performance measure than the usual sample complexity in capturing the asymmetry in the cost of true label realization. Further, since the auditor might request features and labels separately, it is not clear what should count towards sample complexity. To this end, the cost definition provides a unified measure of performance of fairness audit algorithms in the partial feedback setting. Therefore, with this novel definition of audit cost, we now state the main question that we study in this work:

*Design an algorithm that succeeds in the $(\varepsilon, \delta)$-fairness audit with as small audit cost as possible.*

In this work, we study this question under two models of data distribution—*blackbox* and *mixture*. In the blackbox model, no assumption about the data distribution is made. In the mixture model, the distribution of $(X, Y)$ conditioned on $A$, is assumed to come from a mixture of exponential family distributions. Recall that an exponential family distribution over covariate $Z$ (domain $\mathcal{Z}$) and with parameter $\theta \in \mathbb{R}^d$ is given by the probability density function $\mathcal{E}_\theta(Z) := h(Z) \exp(\theta^\mathsf{T} T(Z) - W(\theta))$, where $h : \mathcal{Z} \to \mathbb{R}$ is the base measure, $T : \mathcal{Z} \to \mathbb{R}^d$ is the sufficient statistics and $W : \mathbb{R}^d \to \mathbb{R}$ is the log-partition function. With different choices of $h, T$ and $W$, one can capture many different distributions in this class, notably, multivariate Gaussian, Poisson, Exponential and Binomial.

In our mixture model, the class of distribution from the exponential family is known (*i.e.*, $h, T$ and $W$ are known). The data $(X, A, Y)$ is generated is as follows: first a group label $A$ is sampled, then true label $Y$ is sampled as $Y \sim Ber(q_{1|A})$ and finally, feature $X$ is sampled as $\mathbb{P}[X \mid Y, A] = \mathcal{E}_{\theta^*_{Y,A}}(X) = h(X) \exp(\theta^{*\mathsf{T}}_{Y,A} T(X) - W(\theta^*_{Y,A}))$, where $\theta^*_{y,a} \in \Theta \subset \mathbb{R}^d$, $y \in \{0, 1\}$, $a \in \mathcal{A}$, are unknown. Here $\Theta$ is a known convex set.

In the following section, we highlight our main results. First we bound the audit cost of a natural baseline algorithm for the blackbox model. We then state another algorithm for this setting that has even lower audit cost. Moreover, we also show that this algorithm is tight in terms of the number of true labels it requests by proving a lower bound. Finally, we give another novel algorithm for the mixture model that achieves significantly lower audit cost than the blackbox case. This algorithm has two main novelties. Firstly, it views the past database as truncated samples from the true distribution thus leveraging techniques from learning from truncated samples literature [DGTZ18, LWZ23]. Secondly, it generalizes existing work on learning mixture of distributions from spherical Gaussians [Mah11] to general exponential family.

## 3 Our Results

Let us first define certain quantities that feature in our guarantees. We denote by $\beta := \mathbb{P}[f = 0], \beta_a := \mathbb{P}[f = 0 \mid A = a], p_{y,a} := \mathbb{P}[f = 1, Y = y, A = a], q_{y,a} := \mathbb{P}[Y = y, A = a], p_{y|a} := \mathbb{P}[f = 1, Y = y \mid A = a], q_{y|a} := \mathbb{P}[Y = y \mid A = a]$ for all $y \in \{0, 1\}$ and $a \in \mathcal{A}$, and $\gamma_{ij} := \mathbb{P}[f = i, Y = j]$ for $i, j \in \{0, 1\}$. Note that $p_{y,a} = p_{y|a} \cdot \mathbb{P}[A = a]$, therefore $p_{y,a} \leq p_{y|a}$, and similarly, $q_{y,a} \leq q_{y|a}$. Next, we state the key results of this work.

Recall that due to partial feedback, true labels for negatively classified individuals are not observed. Hence, a natural baseline audit policy is to request true label for every individual with $f = 0$. We have the following characterization of the audit cost of this policy (proof in Appendix A.1).

**Theorem 1.** *Given $\varepsilon, \delta \in (0, 1)$, under the blackbox model, the audit policy that requests true labels of every individual with $f = 0$ (Algorithm 3 in Appendix A) succeeds in the $(\varepsilon, \delta)$-fairness audit with*

$\text{cost} \leq \widetilde{O}\left(\sum_{a\in\mathcal{A}}\sum_{y\in\{0,1\}}\frac{c_{feat}\beta+c_{lab}\gamma_{00}}{q_{y,a}\varepsilon^2}\right)$ . *Moreover, the number of true labels requested by this policy is at most* $\widetilde{O}(\sum_{a\in\mathcal{A}}\sum_{y\in\{0,1\}}\frac{\beta}{q_{y,a}\varepsilon^2})$.

Next, we give an algorithm `RS-Audit` (Algorithm 1) that has improved audit cost using rejection sampling. This algorithm requests an optimal number of true labels, indicating that characterization of the blackbox case is essentially complete. In the design of `RS-Audit` (as well as the natural baseline), we use a threshold-based stopping time construction to decide when to stop requesting true labels. The analysis requires concentration properties of the negative binomial distribution, instead of the usual Chernoff bounds (Lemma 28). Below we formally state the guarantees (proof in A.2).

**Theorem 2.** *For $\varepsilon, \delta \in (0,1)$, in the blackbox model, Algorithm 1 (`RS-Audit`) succeeds in $(\varepsilon, \delta)$-fairness audit with* $\text{cost} \leq \widetilde{O}\left(\sum_{a\in\mathcal{A}}\sum_{y\in\{0,1\}}\frac{c_{feat}\beta_a+c_{lab}(q_{0|a}-p_{0|a})}{q_{y|a}\varepsilon^2}\right)$ . *Moreover, the number of true labels requested by this policy is at most* $\widetilde{O}(\sum_{a\in\mathcal{A}}\sum_{y\in\{0,1\}}\frac{\beta_a}{q_{y|a}\varepsilon^2})$.

The audit cost and the number of true labels requested by `RS-Audit` is lower than the baseline. Note that $\frac{\beta_a}{q_{y|a}} = \frac{\mathbb{P}[f=0,A=a]}{q_{y,a}} \leq \frac{\beta}{q_{y,a}}$. Therefore, the number of true labels requested by the baseline is more than `RS-Audit` by $\widetilde{O}(\sum_{y\in\{0,1\},a\in\mathcal{A}}\frac{\mathbb{P}[f=0,A\neq a]}{q_{y,a}\varepsilon^2})$. Similarly, $\frac{q_{0|a}-p_{0|a}}{q_{y|a}} = \frac{\mathbb{P}[f=0,Y=0,A=a]}{q_{y,a}} \leq \frac{\gamma_{00}}{q_{y,a}}$, hence the audit cost of the baseline is more than `RS-Audit` by $\widetilde{O}(\sum_{y\in\{0,1\},a\in\mathcal{A}}\frac{c_{feat}\mathbb{P}[f=0,A\neq a]+c_{lab}\mathbb{P}[f=0,Y=0,A\neq a]}{q_{y,a}\varepsilon^2})$. Next, we show a lower bound on the number of true labels requested by any algorithm for auditing (proof in Appendix A.3).

**Theorem 3** (Lower Bound). *Under the blackbox setting, for any $\varepsilon \in (0, 1/4)$ and $\delta \in (0,1)$, there exists a family of instances characterized by $p_{y|a}, q_{y|a}$, and $\beta_a$ for $y \in \{0,1\}$ and $a \in \mathcal{A}$, even with $|\mathcal{A}| = 2$, such that any algorithm with finite expected running time that succeeds in the $(\varepsilon, \delta)$-fairness test must request at least $\widetilde{\Omega}\left(\sum_{a\in\mathcal{A}}\sum_{y\in\{0,1\}}\frac{\beta_a}{q_{y|a}\varepsilon^2}\right)$ true labels in expectation.*

Note that the number of true labels requested by `RS-Audit` (Algorithm 1) is within log factors of the lower bound. This in turn indicates that the characterization of the blackbox model is essentially complete. To enrich our understanding of the fairness audit problem under partial feedback, we consider mixture models under some mild structural assumptions. In the mixture model setting, we make the following assumptions: (i) $\mathbb{P}[f = 1 \mid Y = y, A = a] \geq \alpha > 0$ for some known constant $\alpha$, (ii) $\mathcal{E}_\theta(X)$ is log-concave in $X$, $\kappa\mathbf{I} \preceq \nabla^2 W(\theta) \preceq \lambda\mathbf{I}$ with $0 < \kappa < \lambda$ and $W$ is $L$-Lipschitz for all $\theta \in \Theta$, $T(X)$ has components polynomial in $X$, $\mathcal{B}(\theta_{y,a}^*, \frac{1}{\beta}) \subset \Theta$[1] for some $\beta > 0$ for all $y \in \{0,1\}$ and $a \in \mathcal{A}$. Without loss of generality, let $\kappa \leq 1$ and $L, \beta \geq 1$. All these assumptions except Lipschitzness are from [LWZ23] which are used to recover model parameters from truncated data. The positivity assumption (i) also appears in other partial feedback works like [KMC24] and ensures that enough probability mass lies on the positive classification, allowing the distribution recovery. The assumptions in (ii) ensure computational efficiency and concentration bounds. These assumptions along with Lipschitzness are mild and are satisfied by several popular distributions like Gaussian, Exponential and continuous Bernoulli for bounded parameter sets (see Appendix B.4).

Finally, letting $q_{M,a} = q_{1|a} \vee q_{0|a}$ and $q_{m,a} = q_{1|a} \wedge q_{0|a}$, we assume that for all $a \in \mathcal{A}$:

$$\|\theta_{1,a}^* - \theta_{0,a}^*\| \geq \sqrt{\frac{48(L \vee \beta)\beta + 3\lambda}{4\kappa\beta^2}\log\left(\frac{10q_{M,a}}{q_{m,a}\varepsilon}\right)} \tag{1}$$

This separation assumption between parameters enables an approximate MAP based oracle to learn the mixture weights without requesting labels. Such an assumption has been considered in the context of mixture of spherical Gaussians in [Mah11] and has been shown to be mild (and tight) condition for learning good classifiers (see their Table 1 & Remark 8).

Given these aforementioned assumptions, we state our next result which provides an auditing algorithm and its cost bound for the structured setting of the mixture model (proof in Appendix B).

**Theorem 4.** *Given $\varepsilon, \delta \in (0,1)$, under the mixture model and the aforementioned assumptions, `Exp-Audit` (Algorithm 2) succeeds in $(\varepsilon, \delta)$-fairness audit with $\text{cost} \leq$ $\widetilde{O}\left(\sum_{a\in\mathcal{A}}\left(\frac{c_{feat}\beta_a}{q_{m,a}\varepsilon^2} + \sum_{y\in\{0,1\}}\frac{c_{lab}(q_{0|a}-p_{0|a})}{q_{y|a}}\right)\right)$.*

---

[1]This assumption is made in [LWZ23] (see their Claim 1) but not explicitly stated.

The above theorem shows that under structural assumptions on the data distribution, algorithm with better audit cost exists. Notably, the cost concerning $c_{lab}$ is a problem-dependent constant and independent of $\varepsilon$. Our algorithm consists of two key components. First is to view the historical partial feedback data as a collection of truncated samples, thereby applying algorithms for learning distribution parameters from truncated samples [LWZ23]. Second is learning mixture weights with approximate MAP oracle for mixture of exponential family. We believe the first component to be a conceptual contribution since it motivates studying partial feedback, an important setting in fairness in decision-making, through the lens of truncated samples. For the second component, our contribution is a generalization of [Mah11] from spherical Gaussians to exponential family, which we believe could be of independent interest.

In the next sections, we describe RS-Audit and Exp-Audit. Then, we empirically validate our algorithms on the Adult Income [BK96] and Law School [WRC98] datasets.

## 4 Fairness Audit under Blackbox Model

In this section we describe the algorithms for fairness audit under partial feedback for the blackbox model. We first describe the working of the natural baseline policy (details in Algorithm 3, Appendix A) and then another algorithm RS-Audit (Algorithm 1) which improves upon the audit cost and sample complexity of the baseline policy.

Recall that in the audit model under partial feedback, the auditor has to request the true label $Y$ of a negatively classified individual by paying a cost. For the auditor to succeed in fairness audit, it suffices to estimate $\mathbb{P}[f = 1 \mid Y = y, A = a]$ for all $y \in \{0,1\}$ and $a \in \mathcal{A}$. For a given $a \in \mathcal{A}$ and $y \in \{0,1\}$, $\mathbb{P}[f = 1 \mid Y = y, A = a] = \frac{\mathbb{P}[f=1, Y=y, A=a]}{\mathbb{P}[Y=y, A=a]} = \frac{p_{y,a}}{q_{y,a}}$. First, note that $p_{y,a}$ can be estimated from past database using a simple empirical estimate $\frac{1}{n} \sum_{i=1}^{n} \mathbb{1}\{f_i = 1, Y_i = y, A_i = a\}$ without incurring any cost. As the size of the database $n$ grows, this estimate becomes more accurate. Henceforth, we will focus on estimation of $q_{y,a}$ which requires online data, although we will specify how large $n$ is sufficient for the fairness audit.

**Baseline (Algorithm 3).** To estimate $q_{y,a}$, a natural policy is to observe the true labels of all negatively classified individuals. To correctly identify the true hypothesis, it is sufficient to estimate $p_{y,a}$ and $q_{y,a}$ within a multiplicative factor of $1 \pm \frac{\varepsilon}{12}$ for all $y \in \{0,1\}$ and $a \in \mathcal{A}$ (see proof of Theorem 1, Eq. (2) in Appendix A.1). For a fixed $y \in \{0,1\}$ and $a \in \mathcal{A}$, $\widetilde{O}(\frac{1}{p_{y,a}\varepsilon^2})$ and $\widetilde{O}(\frac{1}{q_{y,a}\varepsilon^2})$ samples are sufficient, respectively, via Chernoff bound (Lemma 28). Therefore, the auditor can stop after observing $\widetilde{O}(\frac{1}{q_{y,a}\varepsilon^2})$ true labels from online data (including $f = 1$ individuals and those the auditor requests). However, this insight does not translate into an algorithm since $p_{y,a}$ and $q_{y,a}$ are unknown.

Hence, we base our algorithm on a random stopping time that depends on an appropriately chosen threshold $\tau$ which is a function of $\varepsilon$, $\delta$ and $|\mathcal{A}|$ only. The stopping time for $q_{y,a}$ estimation is defined as $N_{y,a} := \min\{t \geq 1 : \sum_{i=1}^{t} \mathbb{1}\{Y_i = y, A_i = a\} \geq \tau\}$ and likewise define $N'_{y,a}$ for $p_{y,a}$ but with $\mathbb{1}\{f_i = 1, Y_i = y, A_i = a\}$. In other words, for every $y \in \{0,1\}$ and $a \in \mathcal{A}$, the auditor keeps requesting true labels of individuals till a total (including $f = 1$ individuals) of $\tau$ individuals with $Y = y$ and $A = a$ are observed. For a $y \in \{0,1\}$ and $a \in \mathcal{A}$, we set $\widehat{q}_{y,a} = \frac{\tau}{N_{ya}}$ and $\widehat{p}_{y,a} = \frac{\tau}{N'_{y,a}}$ as an estimate of $q_{y,a}$ and $p_{y,a}$ respectively. Using concentration of negative binomials (Lemma 23), we show that these are within $1 \pm \frac{\varepsilon}{12}$ factor of $q_{y,a}$ and $p_{y,a}$. The auditor then computes $\widehat{\Delta} = \max_{y \in \{0,1\}, a, a' \in \mathcal{A}} |\frac{\widehat{p}_{y,a}}{\widehat{q}_{y,a}} - \frac{\widehat{p}_{y,a'}}{\widehat{q}_{y,a'}}|$ and concludes UNFAIR if $\widehat{\Delta} > \frac{\varepsilon}{2}$ and FAIR otherwise.

### 4.1 RS-Audit: Rejection Sampling based Audit

We start with the observation that $\mathbb{P}[f = 1 \mid Y = y, A = a]$ can also be written as:

$$\mathbb{P}[f = 1 \mid Y = y, A = a] = \frac{\mathbb{P}[f = 1, Y = y \mid A = a]}{\mathbb{P}[Y = y \mid A = a]} = \frac{p_{y|a}}{q_{y|a}} .$$

The key insight from this expression is that it suffices to estimate the conditional quantities $p_{y|a}$ and $q_{y|a}$ instead of the joint probabilities $p_{y,a}$ and $q_{y,a}$. With this, we describe RS-Audit (Algorithm 1). For a fixed $a \in \mathcal{A}$, RS-Audit overlooks all samples with $A \neq a$, hence simulating samples from

---

**Algorithm 1** `RS-Audit`: Fairnes Audit Algorithm for Blackbox Model

---

1: Input $\varepsilon, \delta \in (0,1)$ and past database $D$, then set $\tau = 576\log(8|\mathcal{A}|/\delta)/\varepsilon^2$ and $\widehat{\Delta} = 0$.
2: **for** $y \in \{0,1\}$ **do**
3:     **for** $a \in \mathcal{A}$ **do**
4:        $N \leftarrow \texttt{OnlineSample}(\tau, y, a)$, $N' \leftarrow \texttt{PastSample}(D, \tau, y, a)$.
5:        Set $\widehat{q}_{y|a} = \frac{\tau}{N}$ and $\widehat{p}_{y|a} = \frac{\tau}{N'}$. Update $\widehat{\Delta} \leftarrow \max\left\{\widehat{\Delta}, \left|\frac{\widehat{p}_{y|a}}{\widehat{q}_{y|a}} - \frac{\widehat{p}_{y|a'}}{\widehat{q}_{y|a'}}\right|\right\}$
6: **if** $\widehat{\Delta} > \frac{\varepsilon}{2}$ **then** return UNFAIR **else** return FAIR.

---

distribution conditioned on $A = a$ via rejection sampling. In this distribution conditioned on $A = a$, for a $y \in \{0,1\}$, `RS-Audit` proceeds to estimate $q_{y|a}$ via a similar stopping time construction: $N_{y|a} = \min\{t_a \geq 1 : \sum_{i=1}^{t_a} \mathbb{1}\{Y_i = y\} \geq \tau\}$, where $t_a$ denotes the number of individuals with $A = a$. This is done via the subroutine $\texttt{OnlineSample}(\tau, y, a)$ that returns the number of true labels observed (including both $f = 1$ and $f = 0$ individuals) from the distribution conditioned on $A = a$ till $\tau$ individuals with label $Y = y$ are observed. Similarly, $p_{y|a}$ is estimated via subroutine $\texttt{PastSample}(D, \tau, y, a)$ from past database $D$ after removing all individuals with $A \neq a$ to obtain $D_a$ and then using similar stopping time $N'_{y|a}$ but with $\mathbb{1}\{f_i = 1, Y_i = y\}$ on $D_a$, *i.e.*, counting the data points in $D_a$ sequentially till $\tau$ individuals with $\{f = 1, Y = y\}$ are observed. The estimates $\widehat{q}_{y|a} = \frac{\tau}{N_{y|a}}$ and $\widehat{p}_{y|a} = \frac{\tau}{N'_{y|a}}$ are within a factor of $(1 \pm \frac{\varepsilon}{12})$ of their true values (proved using concentration of negative binomials (Lemma 23)). Finally, the algorithm computes $\widehat{\Delta} = \max_{y \in \{0,1\}, a, a' \in \mathcal{A}} |\frac{\widehat{p}_{y|a}}{\widehat{q}_{y|a}} - \frac{\widehat{p}_{y|a'}}{\widehat{q}_{y|a'}}|$ and concludes UNFAIR if $\widehat{\Delta} > \frac{\varepsilon}{2}$ and FAIR otherwise.

*Remark* 1. Although `RS-Audit` and baseline (Algorithm 3) are superficially similar, these are two conceptually different algorithms. The novel insight of not observing the true labels for individuals with $A \neq a$ when trying to estimate $q_{y|a}$ exploits the structure of the audit cost under partial feedback where cost accrues only when the auditor requests the true label. Moreover, we crucially identify the right amount of information needed to estimate $\mathbb{P}[f = 1 \mid Y = y, A = a]$, namely, samples of true labels from the distribution $\mathbb{P}[Y \mid A = a]$. Combining these two observations lead to `RS-Audit`, which is nearly optimal in sample complexity (see Theorem 3). Moreover, due to these observations, the gap in sample complexity of baseline over `RS-Audit` is roughly $\widetilde{O}\left(\sum_{y \in \{0,1\}, a \in \mathcal{A}} \frac{\mathbb{P}[f=0, A \neq a]}{q_{y,a}\varepsilon^2}\right)$ and the difference in audit cost is roughly $\widetilde{O}\left(\sum_{y \in \{0,1\}, a \in \mathcal{A}} \frac{c_{feat}\mathbb{P}[f=0, A \neq a] + c_{lab}\mathbb{P}[Y=0, f=0, A \neq a]}{q_{y,a}\varepsilon^2}\right)$.

## 5 Fairness Audit under Mixture Model

Recall that in the mixture model, given $A = a$, the true label is sampled according to $Y \sim Ber(q_{1|a})$ and given both $A = a$ and $Y = y$, the feature of the individual is sampled according to $X \sim \mathcal{E}_{\theta^*_{y,a}}$ (see Section 2 and 3). Also, we denote $\mathbf{c} := (L, \beta, \kappa, \alpha)$, which are some known quantities of the problem. Now we describe `Exp-Audit` (Algorithm 2) that solves fairness audit under partial feedback in this model with a significantly lower audit cost than the blackbox case.

`Exp-Audit` first splits the past database of positively classified individuals into subsets $D_{y,a}$, $a \in \mathcal{A}$, $y \in \{0,1\}$, so that $D_{y,a}$ only contains individuals with $A = a$ and $Y = y$. Note that samples in $D_{y,a}$ are iid samples of $X \sim \mathcal{E}_{\theta^*_{y,a}}$ truncated by $f$, *i.e.*, $D_{y,a}$ contains only those samples from $\mathcal{E}_{\theta^*_{y,a}}$ for whom $f = 1$. On every subset $D_{y,a}$, `Exp-Audit` applies the subroutine $\texttt{TruncEst}(D_{y,a}, f, \frac{\varepsilon}{2\beta}, \frac{\delta}{14|\mathcal{A}|}, \mathbf{c}, \Theta)$ (an adaptation of [LWZ23], described in Appendix B.2). From $D_{y,a}$ this subroutine learns an estimate $\widehat{\theta}_{y,a}$ of $\theta^*_{y,a}$ such that $\|\widehat{\theta}_{y,a} - \theta^*_{y,a}\| \leq \frac{\varepsilon}{2\beta}$ with probability at least $1 - \frac{\delta}{14|\mathcal{A}|}$ if $|D_{y,a}| \geq \widetilde{\Omega}(\frac{d}{\varepsilon^2})$ (which can be assumed true without loss of generality[2]). Next, the algorithm uses the rejection-sampling based approach similar to `RS-Audit` (line 5) to obtain $\widehat{p}_{y|a} \in (1 \pm \frac{\varepsilon}{12})p_{y|a}$ and $\widetilde{q}_{y|a} \in (1 \pm \varepsilon')q_{y|a}$, where $\varepsilon' = \frac{1}{10}$ is a constant. Note that the cost incurred

---

[2]The auditor can wait without any extra audit cost till this condition is met which happens with high probability in finite time due to the positivity assumption $\mathbb{P}[f = 1 \mid Y = y, A = a] > 0$.

---

**Algorithm 2** `Exp-Audit`: Fairness Audit Algorithm for Mixture Model

---

**Require:** Input $\varepsilon, \delta \in (0,1)$, past database $D$, classifier $f$, $\mathbf{c} = (L, \beta, \kappa, \alpha)$, parameter set $\Theta$.

1: Set $\varepsilon' = 1/10$, $\tau' = 4\log(14|\mathcal{A}|/\delta)/\varepsilon'^2$, $\tau = 576\log(14|\mathcal{A}|/\delta)/\varepsilon^2$ and $\widehat{\Delta} = 0$.
2: **for** $a \in \mathcal{A}$ **do**
3:     **for** $y \in \{0,1\}$ **do**
4:         Set $\widehat{\theta}_{y,a} = \texttt{TruncEst}(D_{y,a}, f, \frac{\varepsilon}{2\beta}, \frac{\delta}{14|\mathcal{A}|}, \mathbf{c}, \Theta)$.
5:         $N' \leftarrow \texttt{OnlineSample}(\tau', y, a)$, $N \leftarrow \texttt{PastSample}(D, \tau, y, a)$
6:         Set $\widetilde{q}_{y|a} = \frac{\tau'}{N'}$ and $\widehat{p}_{y|a} = \frac{\tau}{N}$.
7:     Construct MAP oracle: $M_a(x) = \text{argmax}_{y \in \{0,1\}} \widetilde{q}_{y|a} \mathcal{E}_{\widehat{\theta}_{y,a}}(x)$.
8:     Let $\widetilde{q}_{m,a} = \widetilde{q}_{1|a} \wedge \widetilde{q}_{0|a}$, $R_0 = R_1 = 0$. Observe $R = \frac{3430\log(12|\mathcal{A}|/\delta)}{\widetilde{q}_{m,a}\varepsilon^2}$ features $X$ with $A = a$ and set $R_{M_a(X)} \leftarrow R_{M_a(X)} + 1$ for each $X$.
9:     Set $\widehat{q}_{y|a} = R_y/R$ and update $\widehat{\Delta} \leftarrow \max\left\{\widehat{\Delta}, \left|\frac{\widehat{p}_{y|a}}{\widehat{q}_{y|a}} - \frac{\widehat{p}_{y|a'}}{\widehat{q}_{y|a'}}\right|\right\}$
10: **if** $\widehat{\Delta} > \frac{\varepsilon}{2}$ **then** return UNFAIR **else** return FAIR.

---

in estimating $\widetilde{q}_{y,a}$ is therefore $\widetilde{O}(1)$. This is a crucial difference of `Exp-Audit` with `RS-Audit` which estimates $q_{y|a}$ upto a multiplicative error of $1 \pm \varepsilon$ and incurs $\widetilde{O}(\frac{1}{\varepsilon^2})$ cost.

After this, for every $a \in \mathcal{A}$, `Exp-Audit` constructs a maximum *a posteriori* (MAP) oracle for the mixture model $M_a : \mathcal{X} \to \{0,1\}$, $M_a(X) = \arg\max_{y \in \{0,1\}} \widetilde{q}_{y|a} \mathcal{E}_{\widehat{\theta}_{y,a}}(X)$. In other words, for $A = a$ and feature $X$, the MAP oracle returns which is the most likely true label $Y$. Next, for every $a \in \mathcal{A}$, with $\widetilde{q}_{m,a} = \widetilde{q}_{0|a} \wedge \widetilde{q}_{1|a}$, the algorithm observes only the features $X$ for $R = \widetilde{O}(\frac{1}{\widetilde{q}_{m,a}\varepsilon^2})$ individuals from group $A = a$ and uses the MAP oracle $M_a$ to assign a proxy label to each $X$. With $R_y$ as the number of individuals with proxy label $y$, `Exp-Audit` computes $\widehat{q}_{y|a} = \frac{R_y}{R}$. Under the separation condition (1), we have $\widehat{q}_{y|a} \in (1 \pm \frac{\varepsilon}{12})q_{y|a}$ (see Appendix B.1). Using $\widehat{p}_{y|a}$ and $\widehat{q}_{y|a}$, `Exp-Audit` sets $\widehat{\Delta} = \max_{y \in \{0,1\}, a,a' \in \mathcal{A}} \left|\frac{\widehat{p}_{y|a}}{\widehat{q}_{y|a}} - \frac{\widehat{p}_{y|a'}}{\widehat{q}_{y|a'}}\right|$ and returns UNFAIR if $\widehat{\Delta} > \frac{\varepsilon}{2}$, else FAIR.

*Remark* 2. The cost incurred by `Exp-Audit` is significantly lower than that of `RS-Audit` (see Theorems 2 and 4). In the guarantee of `Exp-Audit` (Theorem 4), the term involved with $c_{lab}$, the cost of requesting a negative label ($Y = 0$), is a problem-dependent constant and independent of $\varepsilon$. In most applications, $c_{lab} \gg c_{feat}$ and therefore, `Exp-Audit` shows that it is possible to achieve a significantly smaller cost under suitable assumptions. This motivates the study of scenarios where the audit cost is better than that for the black-box case, as an interesting future direction.

*Remark* 3. The design of `Exp-Audit` has two novel aspects. The first is to view the past database under partial feedback as a collection of truncated samples from a distribution which allows us to use the past data for learning the distribution parameters using techniques for learning from truncated samples [LWZ23, DGTZ18]. The second is to use MAP oracles as a proxy for true labels under a suitable separation condition (which is weaker than that required for recovering a good clustering [Das99, DS07, VW02, KSV05], see [Mah11] Table 1). To do this, we generalize the result on learning with MAP oracles from spherical Gaussians [Mah11] to general exponential family which may be of independent interest and show that under the novel cost model introduced in this work, it leads to direct improvement in the cost incurred by the algorithm.

*Remark* 4. In `Exp-Audit`, besides solving the fairness audit problem, we also recover the parameters $q_{y|a}$ and $\theta^*_{y,a}$ within $\varepsilon$ accuracy. With these parameters, one can recover a classifier whose error rate is at most $O(\varepsilon)$ more than the error rate of the Bayes optimal classifier. This also allows us to study the fairness-accuracy tradeoff since we can now express the Bayes error and unfairness metrics using distribution parameters with high accuracy [CDPF+17, DWY+20, ZG22].

## 6 Experiments & Results

In the interest of space, a more exhaustive set of experiments are presented in Appendix D, with some key results here. The code can be found at https://github.com/nirjhar-das/fairness_audit_partial_feedback.

**Blackbox Model:** We compare the Baseline (Algorithm 3) and `RS-Audit` algorithms on the Adult Income [BK96] and Law School Admission [WRC98] datasets (details in Appendix D). For a

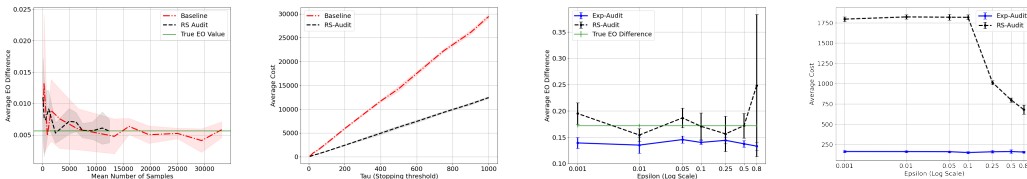

Figure 1: From left, the first two plots show the comparison of the Baseline (Algorithm 3, Appendix A) and the `RS-Audit` (Algorithm 1) on the Adult Income dataset. The last two plots show the comparison of `RS-Audit` and `Exp-Audit` (Algorithm 2).

particular dataset, we first train a classifier on a portion of the dataset. Thereafter, we run our algorithms with this classifier. We simulate past database using a portion of the dataset that is classified positively by the trained classifier, and online samples by drawing random data points from the dataset. We repeat the experiments over 5 seeds and plot the average and the standard deviations. The details of the experimental set up are presented in Appendix D. In our experiments, we vary the stopping threshold $\tau$ over a range and study the behavior of the two algorithms. For every $\tau$, we obtain the mean number of samples, the mean predicted EOD and the mean cost. In Fig. 1 (first and second plots from left), we show the results for a logistic regression classifier trained on all the features on the Adult Dataset. The first plot shows the predicted EOD vs average number of samples collected by the two algorithms. We observe that `RS-Audit` is able to estimate the true EOD (horizontal solid line) with much less samples than Baseline. The second plot shows that `RS-Audit`'s cost is significantly smaller than Baseline at various values of $\tau$.

**Mixture Model:** In the mixture model, we perform synthetic experiments. We choose spherical Gaussians as our exponential family and ensure the separation condition (1) during construction. We generate samples from this mixture model and first train a logistic regression classifier. Thereafter, we simulate past data by collecting some samples on which the classifier's prediction is positive. Since we have the parameters of the Gaussian and the mixture weights, the online data is easily simulated. We experiment with varying values of $\varepsilon$ (with $\delta = 0.01$) and repeat the experiments over 5 seeds. In Fig. 1 (third and fourth plot from left), we show the comparison between `RS-Audit` and `Exp-Audit` (Algorithm 2). The third plot shows mean EOD predicted by the two algorithms and the true EOD (solid green line) at different values of $\varepsilon$. We see that the EOD prediced by `RS-Audit` is closer to the true EOD but has a high standard deviation. On the other hand, `Exp-Audit` underestimates the true EOD by about $0.05$ but has significantly lower standard deviation. For the fourth plot, we set $c_{feat} = 0$ and $c_{lab} = 1$ (other values shown in Appendix D) and we show the mean audit cost of the two algorithms at different values of $\varepsilon$. We observe that the audit cost of `Exp-Audit` is significantly smaller than the audit cost of `RS-Audit`. Moreover, the trend emerges that the cost of `Exp-Audit` is independent of $\varepsilon$ while the cost of `RS-Audit` increases with $\varepsilon$ (note that we had to cap $\tau$ for `RS-Audit` to a maximum of $1000$ for timely execution of the code, hence flattening of the curve).

## 7 Conclusion and Future Work

In this work we studied the fairness audit problem under the partial feedback model. We introduced a novel cost model to capture real-world concerns of exploring additional data. With this cost model, we studied a non-parametric (blackbox) and a mixture of exponential family settings for data distributions. Our framework also combined historical partial feedback data along with online samples. We provided a complete characterization of the blackbox setting with matching upper and lower bounds upto logarithmic factors. In the mixture model setting, we leveraged techniques from learning with truncated samples and MAP oracles to provide an algorithm with significantly lower cost than the blackbox case. We complement our theoretical findings with empirical results that demonstrate the cost efficiency of our proposed algorithms. In this work, we established the connection between partial feedback model with learning from truncated samples which we believe to be a promising research direction. Another interesting future work would be to explore the audit cost complexity of unknown classifiers from a known hypothesis class, e.g., threshold classifiers. Moreover, one can also ask what are other data assumptions under which algorithms with lower audit cost can be designed.

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

# Cost Efficient Fairness Audit Under Partial Feedback: Appendix

## A  Appendix for Blackbox Model

Recall that the quantities of interest are $\beta := \mathbb{P}[f = 0], \beta_a := \mathbb{P}[f = 0 \mid A = a], p_{y,a} := \mathbb{P}[f = 1, Y = y, A = a], q_{y,a} := \mathbb{P}[Y = y, A = a], \gamma_{ij} := \mathbb{P}[f = i, Y = j]$ for $i, j \in \{0, 1\}, p_{y|a} := \mathbb{P}[f = 1, Y = y \mid A = a]$ and $q_{y|a} := \mathbb{P}[Y = y \mid A = a]$ for all $y \in \{0, 1\}$ and $a \in \mathcal{A}$. Note that $p_{y,a} = p_{y|a} \cdot \mathbb{P}[A = a]$, therefore $p_{y,a} \leq p_{y|a}$, and similarly, $q_{y,a} \leq q_{y|a}$. We also have $p_{0,a} \leq p_0$ for all $a \in \mathcal{A}$.

### A.1  Warm-up: Baseline Audit Algorithm for Blackbox Model

We first present the natural baseline algorithm for fairness audit in the blackbox model. Then we give the proof of Theorem 1 which characterizes the audit cost of the algorithm.

---

**Algorithm 3** Naive Audit Algorithm

---

**Require:** Input $\varepsilon, \delta \in (0, 1)$, past database $D$
 1: Set $\tau = 576 \log(8|\mathcal{A}|/\delta)/\varepsilon^2$
 2: **for** $y \in \{0, 1\}$ **do**
 3:   **for** $a \in \mathcal{A}$ **do**
 4:     Observe the true labels $Y$ of individuals till $\tau$ individuals with $Y = y$ and $A = a$ have been observed (including individuals with $f = 1$). Let the number of individuals whose true labels are observed be $N$.
 5:     Iterate over the past database $D$ of the decision-maker till $\tau$ individuals with $f = 1, Y = y$ and $A = a$ are observed. Let the number of individuals iterated over be $N'$.
 6:     Set $\widehat{q}_{y,a} = \frac{\tau}{N}$ and $\widehat{p}_{y,a} = \frac{\tau}{N'}$.
 7: Set $\widehat{\Delta} = \max_{y \in \{0,1\}} \max_{a,a' \in \mathcal{A}} \left| \frac{\widehat{p}_{y,a}}{\widehat{q}_{y,a}} - \frac{\widehat{p}_{y,a'}}{\widehat{q}_{y,a'}} \right|$.
 8: **if** $\widehat{\Delta} > \frac{\varepsilon}{2}$ **then** return UNFAIR **else** return FAIR.

---

The algorithm essentially observes the true labels of all individuals who are negatively classified by the individuals. For a fixed $a \in \mathcal{A}$ and $y \in \{0, 1\}$, the algorithm keeps observing the true labels of individuals till $\tau = O\left(\frac{\log(1/\delta)}{\varepsilon^2}\right)$ individuals of group $a$ with true label $y$ have been observed. If the algorithm stops after observing $N$ individuals for this $a$ and $y$, then the estimate of $q_{y,a}$ is obtained as $\widehat{q}_{y,a} = \frac{\tau}{N}$. To obtain an estimate of $p_{y,a}$, instead of observing true labels over online samples, the algorithm iterates over the past database $D$ till $\tau$ individuals with $f = 1, A = a$ and $Y = y$ are observed. Note that under partial feedback, the true label $Y$ is available to the algorithm for individuals with $f = 1$. Let $N'$ be the number of iterations made by the algorithm in this phase. Then the algorithm sets $\widehat{p}_{y,a} = \frac{\tau}{N'}$. Hereon, the hypothesis testing framework follows in the same way as in Algorithm 1.

### A.1.1  Proof of Theorem 1

First we restate Theorem 1 and then present its proof. Recall that $h : \mathcal{X} \times \mathcal{A} \times \{0, 1\} \to \{0, 1\}$ denotes the policy of the auditor whether to observe the true label of an individual and $g : \mathcal{A} \to \{0, 1\}$ denotes the policy whether to observe the feature of the individual. Note that setting $h = 1$ reveals the features of the individual anyway so when $h = 1$, one can set $g = 0$.

**Theorem 5.** *Given $\varepsilon, \delta \in (0,1)$, under the blackbox model, the audit policy given by $h = 1 - f$, $g = 0$ performs the $(\varepsilon, \delta)$-fairness audit with following audit cost complexity,*

$$\texttt{cost} \leq \tilde{O}\left(\sum_{a \in \mathcal{A}} \sum_{y \in \{0,1\}} \frac{c_{feat}\beta + c_{lab}\gamma_{00}}{q_{y,a}\varepsilon^2}\right).$$

*Moreover, the number of true labels requested by this policy is at most $\tilde{O}(\sum_{a \in \mathcal{A}} \sum_{y \in \{0,1\}} \frac{\beta}{q_{y,a}\varepsilon^2})$.*

*Proof.* Fix an $y \in \{0,1\}$ and an $a \in \mathcal{A}$. The algorithm keeps observing $(A, Y)$ till $\tau = \frac{576 \log(8|\mathcal{A}|/\delta)}{\varepsilon^2}$ samples of $Y = y$ and $A = a$ are observed. Let this number of samples collected be $N_{y,a}$. We define the event $\mathcal{E}_{y,a} := \{N_{y,a} \in (1 \pm \frac{\varepsilon}{12})\frac{576 \log(8|\mathcal{A}|/\delta)}{\varepsilon^2 q_{y,a}}\}$. From Lemma 23, with $\tau = \frac{576 \log(8|\mathcal{A}|/\delta)}{\varepsilon^2}$, we have $\mathbb{P}[\mathcal{E}_{y,a}^c] \leq \frac{\delta}{4|\mathcal{A}|}$. Applying a Union Bound over all $y \in \{0,1\}$ and $a \in \mathcal{A}$, we have for the event $\mathcal{E} := \cap_{(y,a) \in \{0,1\} \times \mathcal{A}} \mathcal{E}_{y,a}$,

$$\mathbb{P}[\mathcal{E}] = \mathbb{P}[\cap_{(a,y) \in \mathcal{A} \times \{0,1\}} \mathcal{E}_{y,a}] = 1 - \mathbb{P}[\cup_{(a,y) \in \mathcal{A} \times \{0,1\}} \mathcal{E}_{y,a}^c] \qquad \text{(De Morgan's law)}$$

$$\geq 1 - \sum_{(a,y) \in \mathcal{A} \times \{0,1\}} \mathbb{P}[\mathcal{E}_{y,a}^c] \qquad \text{(Union Bound)}$$

$$\geq 1 - \sum_{(a,y) \in \mathcal{A} \times \{0,1\}} \frac{\delta}{4|\mathcal{A}|} = 1 - \frac{\delta}{2}.$$

Similarly, we also define the event $\mathcal{F}_{y,a} := \{N'_{y,a} \in (1 \pm \frac{\varepsilon}{12})\frac{576 \log(8|\mathcal{A}|/\delta)}{\varepsilon^2 p_{y,a}}\}$, where $N'_{y,a}$ is the number of individuals iterated over by the algorithm in the past database while trying to collect $\tau$ individuals with $f = 1, Y = y$ and $A = a$. By essentially similar arguments as above (application of Lemma 23), we can show that the event $\mathcal{F} := \cap_{(y,a) \in \{0,1\} \times \mathcal{A}} \mathcal{F}_{y,a}$ happens with probability at least $1 - \frac{\delta}{2}$. Therefore, via Union Bound, $\mathbb{P}[\mathcal{E} \cap \mathcal{F}] \geq 1 - \delta$.

Under event $\mathcal{E}$, we have that the total number of samples collected by the algorithm:

$$N := \sum_{(a,y) \in \mathcal{A} \times \{0,1\}} N_{y,a} \leq \sum_{(a,y) \in \mathcal{A} \times \{0,1\}} \left(1 + \frac{\varepsilon}{12}\right) \frac{576 \log(8|\mathcal{A}|/\delta)}{\varepsilon^2 q_{y,a}}$$

$$\leq \sum_{(a,y) \in \mathcal{A} \times \{0,1\}} \frac{13}{12} \cdot \frac{576 \log(8|\mathcal{A}|/\delta)}{\varepsilon^2 q_{y,a}} \qquad (\varepsilon < 1)$$

$$= \sum_{a \in \mathcal{A}} \sum_{y \in \{0,1\}} \frac{624 \log(8|\mathcal{A}|/\delta)}{\varepsilon^2 q_{y,a}}$$

Finally, the expected number of individuals for whom $h = 1$ is $N \cdot \mathbb{P}[h = 1] = N \cdot \mathbb{P}[f = 0] = N\beta$. This is the actual number of individuals for whom the algorithm requests true labels. Hence, the number of true labels requested by this algorithm is at most

$$N \cdot \beta \leq \sum_{a \in \mathcal{A}} \sum_{y \in \{0,1\}} \frac{624\beta \log(8|\mathcal{A}|/\delta)}{\varepsilon^2 q_{y,a}}.$$

Further, the audit cost of observing the true labels of $N$ individuals is $N \cdot (c_{feat}\mathbb{P}[f = 0] + c_{lab}\mathbb{P}[f = 0, Y = 0]) = N \cdot (c_{feat}\beta + c_{lab}\gamma_{00})$. Hence, the audit cost of the algorithm is bounded by

$$\texttt{cost} \leq \sum_{a \in \mathcal{A}} \sum_{y \in \{0,1\}} \frac{624(c_{feat}\beta + c_{lab}\gamma_{00}) \log(8|\mathcal{A}|/\delta)}{\varepsilon^2 q_{y,a}}.$$

Finally, it remains to show that the algorithm also successfully performs the hypothesis test. Under the event $\mathcal{E} \cap \mathcal{F}$, we have $\widehat{q}_{y,a} = \frac{\tau}{N_{y,a}} = \frac{576 \log(8|\mathcal{A}|/\delta)}{\varepsilon^2 N_{y,a}}$. Hence, $\frac{q_{y,a}}{1+\frac{\varepsilon}{12}} \leq \widehat{q}_{y,a} \leq \frac{q_{y,a}}{1-\frac{\varepsilon}{12}}$. Similarly, $\frac{p_{y,a}}{1+\frac{\varepsilon}{12}} \leq \widehat{p}_{y,a} \leq \frac{p_{y,a}}{1-\frac{\varepsilon}{12}}$. Hence,

$$\frac{(1-\frac{\varepsilon}{12})p_{y,a}}{(1+\frac{\varepsilon}{12})q_{y,a}} \leq \frac{\widehat{p}_{y,a}}{\widehat{q}_{y,a}} \leq \frac{(1+\frac{\varepsilon}{12})p_{y,a}}{(1-\frac{\varepsilon}{12})q_{y,a}} \ . \tag{2}$$

Using Lemma 10 and the fact that $0 \leq \frac{p_{y,a}}{q_{y,a}} \leq 1$, we finally have,

$$\frac{p_{y,a}}{q_{y,a}} - \frac{\varepsilon}{6} \leq \frac{\widehat{p}_{y,a}}{\widehat{q}_{y,a}} \leq \frac{p_{y,a}}{q_{y,a}} + \frac{\varepsilon}{3} \ .$$

Since the above holds for all $(y,a) \in \{0,1\} \times \mathcal{A}$ under $\mathcal{E} \cap \mathcal{F}$, we therefore have, for a particular $y \in \{0,1\}$ and $a, a' \in \mathcal{A}$,

$$\frac{p_{y,a}}{q_{y,a}} - \frac{p_{y,a'}}{q_{y,a'}} - \frac{\varepsilon}{2} \leq \frac{\widehat{p}_{y,a}}{\widehat{q}_{y,a}} - \frac{\widehat{p}_{y,a'}}{\widehat{q}_{y,a'}} \leq \frac{p_{y,a}}{q_{y,a}} - \frac{p_{y,a'}}{q_{y,a'}} + \frac{\varepsilon}{2} \ .$$

From this, we can conclude that

$$\left| \left| \frac{\widehat{p}_{y,a}}{\widehat{q}_{y,a}} - \frac{\widehat{p}_{y,a'}}{\widehat{q}_{y,a'}} \right| - \left| \frac{p_{y,a}}{q_{y,a}} - \frac{p_{y,a'}}{q_{y,a'}} \right| \right| \leq \frac{\varepsilon}{2} \ . \tag{3}$$

Suppose the true hypothesis is UNFAIR. Let $y^* \in \{0,1\}$ and $a_1, a_2 \in \mathcal{A}, a_1 \neq a_2$ be such that $|\mathbb{P}[f = 1 \mid Y = y^*, A = a_1] - \mathbb{P}[f = 1 \mid Y = y^*, A = a_2]| = \left| \frac{p_{y^*,a_1}}{q_{y^*,a_1}} - \frac{p_{y^*,a_2}}{q_{y^*,a_2}} \right| > \varepsilon$. Then, we have,

$$\widehat{\Delta} \geq \left| \frac{\widehat{p}_{y^*,a_1}}{\widehat{q}_{y^*,a_1}} - \frac{\widehat{p}_{y^*,a_2}}{\widehat{q}_{y^*,a_2}} \right| \qquad \text{(Definition of } \widehat{\Delta} \text{ (Step 9 of Algorithm 3))}$$

$$\geq \left| \frac{p_{y^*,a_1}}{q_{y^*,a_1}} - \frac{p_{y^*,a_1}}{q_{y^*,a_2}} \right| - \frac{\varepsilon}{2} \qquad \text{(Equation (3))}$$

$$> \varepsilon - \frac{\varepsilon}{2} \qquad \qquad \left( \left| \frac{p_{y^*,a_1}}{q_{y^*,a_1}} - \frac{p_{y^*,a_2}}{q_{y^*,a_2}} \right| > \varepsilon \right)$$

$$= \frac{\varepsilon}{2} \ .$$

The algorithm therefore outputs UNFAIR (Step 10). Hence, under the hypothesis UNFAIR, the algorithm's output is correct.

Next, suppose the true hypothesis is FAIR. Then, we have for all $y \in \{0,1\}$ and $a, a' \in \mathcal{A}$,

$$|\mathbb{P}[f = 1 \mid Y = y, A = a] - \mathbb{P}[f = 1 \mid Y = y, A = a']| = \left| \frac{p_{y,a}}{q_{y,a}} - \frac{p_{y,a'}}{q_{y,a'}} \right| = 0 \ .$$

Hence, by Equation (3), we can conclude that for all $y \in \{0,1\}$ and $a, a' \in \mathcal{A}$,

$$\left| \frac{\widehat{p}_{y,a}}{\widehat{q}_{y,a}} - \frac{\widehat{p}_{y,a'}}{\widehat{q}_{y,a'}} \right| \leq \left| \frac{p_{y,a}}{q_{y,a}} - \frac{p_{y,a'}}{q_{y,a'}} \right| + \frac{\varepsilon}{2}$$

$$= 0 + \frac{\varepsilon}{2} \qquad \qquad \left( \left| \frac{p_{y,a}}{q_{y,a}} - \frac{p_{y,a'}}{q_{y,a'}} \right| = 0 \right)$$

$$= \frac{\varepsilon}{2} \ .$$

In particular, we have $\widehat{\Delta} = \max_{y \in \{0,1\}} \max_{a,a' \in \mathcal{A}} \left| \frac{\widehat{p}_{y,a}}{\widehat{q}_{y,a}} - \frac{\widehat{p}_{y,a'}}{\widehat{q}_{y,a'}} \right| \leq \frac{\varepsilon}{2}$. Hence, the algorithm outputs FAIR implying that the algorithm's output is correct.

Since the output of the algorithm is correct under both the hypotheses, one can conclude the output of the algorithm is always correct. Note that we assumed event $\mathcal{E} \cap \mathcal{F}$ to hold, under which the correctness of our algorithm was derived. Therefore, our algorithm is correct with probability at least $1 - \delta$. This finishes the proof.

$\square$

## A.2 Performance of `RS-Audit` in Blackbox Model

In this section we present the analysis of `RS-Audit` (Algorithm 1) under the blackbox model. In particular, we state the proof of Theorem 2. We now restate the theorem.

**Theorem 6.** *Given $\varepsilon, \delta \in (0, 1)$, RS-Audit (Algorithm 1) successfully performs the $(\varepsilon, \delta)$-fairness audit with the following audit cost complexity:*

$$\texttt{cost} \leq \tilde{O}\left(\sum_{a \in \mathcal{A}} \sum_{y \in \{0,1\}} \frac{c_{feat}\beta_a + c_{lab}(q_{0|a} - p_{0|a})}{q_{y|a}\varepsilon^2}\right).$$

*Moreover, the number of true labels requested by this policy is at most $\tilde{O}(\sum_{a \in \mathcal{A}} \sum_{y \in \{0,1\}} \frac{\beta_a}{q_{y|a}\varepsilon^2})$.*

*Proof.* The proof essentially follows similar arguments as in the proof of Theorem 5 with the crucial difference that we now consider the conditional distributions $\mathbb{P}[Y \mid A]$. This is because Algorithm 1 iterates over all $y \in \{0, 1\}$ and $a \in \mathcal{A}$ with the key difference that in a particular iteration, *i.e.*, for a fixed $y \in \{0, 1\}$ and $a \in \mathcal{A}$, the algorithm only requests the true labels of individuals with $A = a$. This is akin to collecting samples from the conditional distribution obtained by conditioning on $A = a$. Therefore, the arguments have to be adapted to the conditional distribution.

In particular, let us fix $y \in \{0, 1\}$ and $a \in \mathcal{A}$. Let $N_{y|a}$ be the number of samples collected by the algorithm in the corresponding iteration (Step 4 of the algorithm). Consider the event $\mathcal{E}_{y,a} := \{N_{y|a} \in \left(1 \pm \frac{\varepsilon}{12}\right) \frac{576 \log(8|\mathcal{A}|/\delta)}{\varepsilon^2 q_{y|a}}\}$ (recall that $q_{y|a} = \mathbb{P}[Y = y \mid A = a]$). From Lemma 23, with $\tau = \frac{576 \log(8|\mathcal{A}|/\delta)}{\varepsilon^2}$, we have that $\mathbb{P}[\mathcal{E}_{y,a}^c] \leq \frac{\delta}{4|\mathcal{A}|}$. Therefore, the event $\mathcal{E} := \cap_{(y,a) \in \{0,1\} \times \mathcal{A}} \mathcal{E}_{y,a}$ happens with probability at least $1 - \frac{\delta}{2}$ by Union Bound. We also define $\mathcal{F} := \{N'_{y|a} \in \left(1 \pm \frac{\varepsilon}{12}\right) \frac{576 \log(8|\mathcal{A}|/\delta)}{\varepsilon^2 p_{y|a}}\}$, where $N'_{y|a}$ is the number of samples collected in Step 5 of the algorithm. Once again, by application of Lemma 23 and Union Bound, the event $\mathcal{F} := \cap_{(y,a) \in \{0,1\} \times \mathcal{A}} \mathcal{F}_{y,a}$ happens with probability at least $1 - \frac{\delta}{2}$. Therefore, $\mathbb{P}[\mathcal{E} \cap \mathcal{F}] \geq 1 - \delta$.

Under $\mathcal{E}$, the total number of online samples collected by Algorithm 1 is

$$\begin{aligned} N := \sum_{(y,a) \in \{0,1\} \times \mathcal{A}} N_{y|a} &\leq \sum_{(y,a) \in \{0,1\} \times \mathcal{A}} \left(1 + \frac{\varepsilon}{12}\right) \frac{576 \log(8|\mathcal{A}|/\delta)}{\varepsilon^2 q_{y|a}} \\ &\leq \sum_{(y,a) \in \{0,1\} \times \mathcal{A}} \frac{13}{12} \cdot \frac{576 \log(8|\mathcal{A}|/\delta)}{\varepsilon^2 q_{y|a}} \qquad (\varepsilon < 1) \\ &= \sum_{a \in \mathcal{A}} \sum_{y \in \{0,1\}} \frac{624 \log(8|\mathcal{A}|/\delta)}{\varepsilon^2 q_{y|a}} \, . \end{aligned}$$

Note that $N_{y|a}$ samples are collected by the algorithm under the conditional distribution $\mathbb{P}[Y \mid A = a]$. Therefore, the expected number of samples actually requested by the algorithm is $N_{y|a} \cdot \mathbb{P}[f = 0 \mid A = a] = N_{y|a}\beta_a$. Similarly, the audit cost of $N_{y|a}$ samples is $N_{y|a}(c_{feat}\mathbb{P}[f = 0 \mid A = a] + c_{lab}\mathbb{P}[f = 0, Y = 0 \mid A = a]) = N_{y|a}(c_{feat}\beta_a + c_{lab}(q_{0|a} - p_{0|a}))$. Therefore, the expected number of samples requested by the algorithm is at most

$$\sum_{y \in \{0,1\}} \sum_{a \in \mathcal{A}} N_{y|a} \cdot \beta_a \leq \sum_{a \in \mathcal{A}} \sum_{y \in \{0,1\}} \frac{624 \beta_a \log(8|\mathcal{A}|/\delta)}{\varepsilon^2 q_{y|a}} \, .$$

Similarly, the audit cost of Algorithm 1 can be bounded as

$$\texttt{cost} = \sum_{y \in \{0,1\}} \sum_{a \in \mathcal{A}} N_{y|a}(c_{feat}\beta_a + c_{lab}(q_{0|a} - p_{0|a})) \leq \sum_{a \in \mathcal{A}} \sum_{y \in \{0,1\}} \frac{624(c_{feat}\beta_a + c_{lab}(q_{0|a} - p_{0|a})) \log(8|\mathcal{A}|/\delta)}{\varepsilon^2 q_{y|a}} \, .$$

The correctness of Algorithm 1 can be shown in the same way as in the proof of Theorem 5. This is because under event $\mathcal{E} \cap \mathcal{F}$, we have $\widehat{q}_{y|a} = \frac{\tau}{N_{y|a}} = \frac{576 \log(8|\mathcal{A}|/\delta)}{\varepsilon^2 N_{y|a}}$. Hence, $\frac{q_{y|a}}{1+\frac{\varepsilon}{12}} \leq \widehat{q}_{y|a} \leq \frac{q_{y|a}}{1-\frac{\varepsilon}{12}}$. Similarly, $\frac{p_{y|a}}{1+\frac{\varepsilon}{12}} \leq \widehat{p}_{y|a} \leq \frac{p_{y|a}}{1-\frac{\varepsilon}{12}}$. Hence,

$$\frac{(1-\frac{\varepsilon}{12})p_{y|a}}{(1+\frac{\varepsilon}{12})q_{y|a}} \leq \frac{\widehat{p}_{y|a}}{\widehat{q}_{y|a}} \leq \frac{(1+\frac{\varepsilon}{12})p_{y|a}}{(1-\frac{\varepsilon}{12})q_{y|a}} \ . \tag{4}$$

Using Lemma 10 and the fact that $0 \leq \frac{p_{y|a}}{q_{y|a}} \leq 1$, we finally have,

$$\frac{p_{y|a}}{q_{y|a}} - \frac{\varepsilon}{6} \leq \frac{\widehat{p}_{y|a}}{\widehat{q}_{y|a}} \leq \frac{p_{y|a}}{q_{y|a}} + \frac{\varepsilon}{3} \ .$$

Since the above holds for all $(y, a) \in \{0, 1\} \times \mathcal{A}$ under $\mathcal{E} \cap \mathcal{F}$, we therefore have, for a particular $y \in \{0, 1\}$ and $a, a' \in \mathcal{A}$,

$$\frac{p_{y|a}}{q_{y|a}} - \frac{p_{y|a'}}{q_{y|a'}} - \frac{\varepsilon}{2} \leq \frac{\widehat{p}_{y|a}}{\widehat{q}_{y|a}} - \frac{\widehat{p}_{y|a'}}{\widehat{q}_{y|a'}} \leq \frac{p_{y|a}}{q_{y|a}} - \frac{p_{y|a'}}{q_{y|a'}} + \frac{\varepsilon}{2} \ .$$

From this, we can conclude that

$$\left| \left| \frac{\widehat{p}_{y|a}}{\widehat{q}_{y|a}} - \frac{\widehat{p}_{y|a'}}{\widehat{q}_{y|a'}} \right| - \left| \frac{p_{y|a}}{q_{y|a}} - \frac{p_{y|a'}}{q_{y|a'}} \right| \right| \leq \frac{\varepsilon}{2} \ . \tag{5}$$

Suppose the true hypothesis is UNFAIR. Let $y^* \in \{0, 1\}$ and $a_1, a_2 \in \mathcal{A}, a_1 \neq a_2$ be such that $|\mathbb{P}[f = 1 \mid Y = y^*, A = a_1] - \mathbb{P}[f = 1 \mid Y = y^*, A = a_2]| = \left| \frac{p_{y^*|a_1}}{q_{y^*|a_1}} - \frac{p_{y^*|a_2}}{q_{y^*|a_2}} \right| > \varepsilon$. Then, we have,

$$\widehat{\Delta} \geq \left| \frac{\widehat{p}_{y^*|a_1}}{\widehat{q}_{y^*|a_1}} - \frac{\widehat{p}_{y^*|a_2}}{\widehat{q}_{y^*|a_2}} \right| \qquad \text{(Definition of } \widehat{\Delta} \text{ (Step 9 of Algorithm 3))}$$

$$\geq \left| \frac{p_{y^*|a_1}}{q_{y^*,a_1}} - \frac{p_{y^*|a_1}}{q_{y^*|a_2}} \right| - \frac{\varepsilon}{2} \qquad \text{(Equation (3))}$$

$$> \varepsilon - \frac{\varepsilon}{2} \qquad \left( \left| \frac{p_{y^*|a_1}}{q_{y^*|a_1}} - \frac{p_{y^*|a_2}}{q_{y^*|a_2}} \right| > \varepsilon \right)$$

$$= \frac{\varepsilon}{2} \ .$$

The algorithm therefore outputs UNFAIR (Step 10). Hence, under the hypothesis UNFAIR, the algorithm's output is correct.

Next, suppose the true hypothesis is FAIR. Then, we have for all $y \in \{0, 1\}$ and $a, a' \in \mathcal{A}$,

$$|\mathbb{P}[f = 1 \mid Y = y, A = a] - \mathbb{P}[f = 1 \mid Y = y, A = a']| = \left| \frac{p_{y|a}}{q_{y|a}} - \frac{p_{y|a'}}{q_{y|a'}} \right| = 0 \ .$$

Hence, by Equation (5), we can conclude that for all $y \in \{0, 1\}$ and $a, a' \in \mathcal{A}$,

$$\left| \frac{\widehat{p}_{y|a}}{\widehat{q}_{y|a}} - \frac{\widehat{p}_{y|a'}}{\widehat{q}_{y|a'}} \right| \leq \left| \frac{p_{y|a}}{q_{y|a}} - \frac{p_{y|a'}}{q_{y|a'}} \right| + \frac{\varepsilon}{2}$$

$$= 0 + \frac{\varepsilon}{2} \qquad \left( \left| \frac{p_{y|a}}{q_{y|a}} - \frac{p_{y|a'}}{q_{y|a'}} \right| = 0 \right)$$

$$= \frac{\varepsilon}{2} \ .$$

In particular, we have $\widehat{\Delta} = \max_{y \in \{0,1\}} \max_{a,a' \in \mathcal{A}} \left| \frac{\widehat{p}_{y|a}}{\widehat{q}_{y|a}} - \frac{\widehat{p}_{y|a'}}{\widehat{q}_{y|a'}} \right| \leq \frac{\varepsilon}{2}$. Hence, the algorithm outputs FAIR implying that the algorithm's output is correct.

Since the output of the algorithm is correct under both the hypotheses, one can conclude the output of the algorithm is always correct. Note that we assumed event $\mathcal{E} \cap \mathcal{F}$ to hold, under which the correctness of our algorithm was derived. Therefore, our algorithm is correct with probability at least $1 - \delta$. This finishes the proof. □

## A.3 Lower Bound for the Blackbox Model

**Canonical Partial Feedback Model.** Let us consider the sequential interaction of an algorithm $\pi$ with a partial feedback system given $A = a$, that is, the algorithm only observes samples from the conditional distribution $A = a$. At round $t$, the environment first generates an iid sample $(Y_t, f_t)$, where $Y_t$ is the true label and $f_t$ is the classifiers decision. First, the $f_t$ is visible to the algorithm. If $f_t = 1$, the algorithm takes no action (denote by $x_t = \odot$) on the environment (*i.e.*, only internal computations). Then $Z_t = Y_t$ is revealed to the algorithm. The round ends and the algorithm can update its policy from $\pi_t$ to $\pi_{t+1}$. On the other hand, if $f_t = 0$, the algorithm must decide whether to observe $Y_t$ (action $x_t = 1$) or not (action $x_t = 0$). If $x_t = 1$, the label $Y_t$ is revealed to the algorithm, *i.e.*, $Z_t = Y_t$, else $Z_t = \star$. After this, the algorithm updates its policy to $\pi_{t+1}$.

This gives rise to a natural filtration $\mathcal{F}_t = \sigma(f_1, x_1, Z_1, f_2, x_2, Z_2, \ldots, f_t, x_t, Z_t)$. Let us consider the probability of this random vector $\{f_1, \ldots, Z_t\}$ under some hypothesis $M$ generating the data.

$$
\begin{aligned}
\mathbb{P}_{M\pi}(f_1, x_1, Z_1, \ldots, f_t, x_t, Z_t) &= \prod_{s=1}^{t} \mathbb{P}_{M\pi}[Z_s \mid f_1, \ldots, x_s] \cdot \mathbb{P}_{M\pi}[x_s \mid f_1, x_1, Z_1, \ldots, f_s] \cdot \mathbb{P}_{M\pi}[f_s \mid f_1, \ldots, Z_{s-1}] \\
&= \prod_{s=1}^{t} \mathbb{P}_{M\pi}[Z_s \mid f_s, x_s] \cdot \pi_s(x_s \mid f_1, x_1, Z_1, \ldots, f_s) \cdot \mathbb{P}_M[f_s]
\end{aligned}
$$

Here, we have used the fact that the samples $f_t$'s are iid and $Z_t$ is conditionally independent of the past given $f_t$ and $x_t$, so that $\mathbb{P}_{M\pi}[Z_s \mid f_1, \ldots, x_s] = \mathbb{P}_{M\pi}[Z_s \mid f_s, x_s]$ and $\mathbb{P}_{M\pi}[f_s \mid f_1, \ldots, Z_{s-1}] = \mathbb{P}_M[f_s]$.

Now suppose there is an alternate hypothesis that also generates the data such that $\mathbb{P}_N[f_t] = \mathbb{P}_M[f_t]$. Then, we have

$$
\frac{\mathbb{P}_M(f_1, x_1, Z_1, \ldots, f_t, x_t, Z_t)}{\mathbb{P}_N(f_1, x_1, Z_1, \ldots, f_t, x_t, Z_t)} = \prod_{s=1}^{t} \frac{\mathbb{P}_M[Z_s \mid f_s, x_s] \cdot \pi_s(x_s \mid f_1, \ldots, f_s) \cdot \mathbb{P}_M[f_s]}{\mathbb{P}_N[Z_s \mid f_s, x_s] \cdot \pi_s(x_s \mid f_1, \ldots, f_s) \cdot \mathbb{P}_N[f_s]} = \prod_{s=1}^{t} \frac{\mathbb{P}_M[Z_s \mid f_s, x_s]}{\mathbb{P}_N[Z_s \mid f_s, x_s]} \tag{6}
$$

**Lemma 7** (Divergence Decomposition for Canonical Partial Feedback Model). *Suppose $M$ and $N$ are two hypotheses for the canonical partial feedback model defined above and let $\pi$ be an algorithm interacting with this feedback model till round $t \in \mathbb{N}$. Let $M\pi$ and $N\pi$ denote the probability measures under $M$ and $N$ respectively for the random vector $(f_1, x_1, Z_1, \ldots, f_t, x_t, Z_t)$. If it holds that $\mathbb{P}_N[f_t] = \mathbb{P}_M[f_t]$, then, with $V_t = (f_1, x_1, Z_t, \ldots, Z_t)$*

$$
\begin{aligned}
D_{KL}(\mathbb{P}_{M\pi}[V_t] \parallel \mathbb{P}_{N\pi}[V_t]) = t \cdot \mathbb{P}_M[f = 1] \cdot D_{KL}(\mathbb{P}_M[Y \mid f = 1] \parallel \mathbb{P}_N[Y \mid f = 1]) \\
+ \mathbb{E}_{M\pi}[L_t] \cdot D_{KL}(\mathbb{P}_{M\pi}[Y \mid f = 0] \parallel \mathbb{P}_{N\pi}[Y \mid f = 0]),
\end{aligned}
$$

*where $L_t := \sum_{s=1}^{t} \mathbb{1}\{f_s = 0, x_s = 1\}$.*

*Proof.* We have that

$$D_{KL}(\mathbb{P}_{M\pi}[V_t] \parallel \mathbb{P}_{N\pi}[V_t]) = \mathbb{E}_{M\pi}\left[\log\left(\frac{\mathbb{P}_M(f_1, x_1, Z_1, \ldots, f_t, x_t, Z_t)}{\mathbb{P}_N(f_1, x_1, Z_1, \ldots, f_t, x_t, Z_t)}\right)\right]$$

$$= \mathbb{E}_{M\pi}\left[\sum_{s=1}^t \log\left(\frac{\mathbb{P}_M[Z_s \mid f_s, x_s]}{\mathbb{P}_N[Z_s \mid f_s, x_s]}\right)\right] \qquad \text{(via Equation (6) since } \mathbb{P}_M[f] = \mathbb{P}_M[f])$$

$$= \sum_{s=1}^t \mathbb{E}_{M\pi}\left[\mathbb{E}_{M\pi}\left[\log\left(\frac{\mathbb{P}_M[Z_s \mid f_s, x_s]}{\mathbb{P}_N[Z_s \mid f_s, x_s]}\right)\Big|(f_s, x_s)\right]\right]$$

$$= \sum_{s=1}^t \mathbb{E}_{M\pi}\left[D_{KL}\left(\mathbb{P}_{M\pi}[Z_s \mid f_s, x_s] \parallel \mathbb{P}_{N\pi}[Z_s \mid f_s, x_s]\right)\right]$$

$$= \sum_{s=1}^t \mathbb{E}_{M\pi}\left[\mathbb{1}\{f_s = 1, x_s = \circledcirc\} \cdot D_{KL}\left(\mathbb{P}_{M\pi}[Z_s \mid f_s = 1, x_s = \circledcirc] \parallel \mathbb{P}_{N\pi}[Z_s \mid f_s = 1, x_s = \circledcirc]\right)\right.$$

$$+ \mathbb{1}\{f_s = 0, x_s = 0\} \cdot D_{KL}\left(\mathbb{P}_{M\pi}[Z_s \mid f_s = 0, x_s = 0] \parallel \mathbb{P}_{N\pi}[Z_s \mid f_s = 0, x_s = 0]\right)$$

$$\left.+ \mathbb{1}\{f_s = 0, x_s = 1\} \cdot D_{KL}\left(\mathbb{P}_{M\pi}[Z_s \mid f_s = 0, x_s = 1] \parallel \mathbb{P}_{N\pi}[Z_s \mid f_s = 0, x_s = 1]\right)\right]$$

$$= \sum_{s=1}^t \mathbb{E}_{M\pi}\left[\mathbb{1}\{f_s = 1\} \cdot D_{KL}\left(\mathbb{P}_{M\pi}[Y_s \mid f_s = 1] \parallel \mathbb{P}_{N\pi}[Y_s \mid f_s = 1]\right)\right.$$

$$+ \mathbb{1}\{f_s = 0, x_s = 0\} \cdot D_{KL}\left(\mathbb{P}_{M\pi}[\star \mid f_s = 0, x_s = 0] \parallel \mathbb{P}_{N\pi}[\star \mid f_s = 0, x_s = 0]\right)$$

$$\left.+ \mathbb{1}\{f_s = 0, x_s = 1\} \cdot D_{KL}\left(\mathbb{P}_{M\pi}[Y_s \mid f_s = 0, x_s = 1] \parallel \mathbb{P}_{N\pi}[Y_s \mid f_s = 0, x_s = 1]\right)\right]$$

$$= \sum_{s=1}^t \mathbb{E}_M[\mathbb{1}\{f = 1\}] \cdot D_{KL}\left(\mathbb{P}_M[Y \mid f = 1] \parallel \mathbb{P}_N[Y \mid f = 1]\right)$$

$$+ 0 + \mathbb{E}_{M\pi}\left[\sum_{s=1}^t \mathbb{1}\{f_s = 0, x_s = 1\}\right] \cdot D_{KL}\left(\mathbb{P}_M[Y \mid f = 0] \parallel \mathbb{P}_N[Y \mid f = 0]\right)$$

$$\hfill ((f_s, Y_s) \text{ are iid samples})$$

$$= t \cdot \mathbb{P}_M[f = 1] \cdot D_{KL}\left(\mathbb{P}_M[Y \mid f = 1] \parallel \mathbb{P}_N[Y \mid f = 1]\right)$$

$$+ \mathbb{E}_{M\pi}[L_t] \cdot D_{KL}\left(\mathbb{P}_M[Y \mid f = 0] \parallel \mathbb{P}_N[Y \mid f = 0]\right)$$

$$\hfill \square$$

**Lemma 8** (Divergence Decomposition with Stopping Time)**.** *With notations same as in Lemma 7, let $\pi$ be an algorithm that interacting with the model till a $(\mathcal{F}_t)_{t=1}^\infty$-measurable stopping time $\tau$. Further, let $\mathbb{E}_{M\pi}[\tau] < \infty$. Let $\mathcal{O}$ be $\mathcal{F}_\tau$-measurable random variable. Then,*

$$D_{KL}(\mathbb{P}_{M\pi}[\mathcal{O}] \parallel \mathbb{P}_{N\pi}[\mathcal{O}]) \leq \mathbb{E}_{M\pi}[\tau] \cdot \mathbb{P}_M[f = 1] \cdot D_{KL}(\mathbb{P}_M[Y \mid f = 1] \parallel \mathbb{P}_N[Y \mid f = 1])$$

$$+ \mathbb{E}_{M\pi}[L_\tau] \cdot D_{KL}(\mathbb{P}_{M\pi}[Y \mid f = 0] \parallel \mathbb{P}_{N\pi}[Y \mid f = 0])$$

*where $L_\tau$ is the stopped version of $L_t$.*

*Proof.* The proof can be obtained by combining Lemma 7 with data processing inequality and equation relating KL-divergences between push-forward measures. For reference see Proof of Lemma 2.6 in [Ste23]. $\square$

Now we restate Theorem 3 for the sample complexity lower bound in blackbox model and present its proof.

**Theorem 9.** *Under the blackbox setting, for any $\varepsilon \in (0, 1/4)$ and $\delta \in (0, 1)$, there exists a family of instances characterized by $p_{y|a}$, $q_{y|a}$, and $\beta_a$ for $y \in \{0, 1\}$ and $a \in \mathcal{A}$, even with $|\mathcal{A}| = 2$, such that any algorithm with finite expected running time that succeeds in the $(\varepsilon, \delta)$-fairness test must request at least $\widetilde{\Omega}\left(\sum_{a \in \mathcal{A}} \sum_{y \in \{0,1\}} \frac{\beta_a}{q_{y|a}\varepsilon^2}\right)$ true labels in expectation.*

*Proof.* We will construct two instances with $\mathcal{A} = \{0, 1\}$ such that they are close, yet in one case the classifier is fair and in the other, it is unfair. However since the instances are close, any algorithm using less than a certain number of samples will not be able to distinguish between the two.

Let $p, q \in (0, 1)$ be constants to be decided later. The classifier $f$ is given as follows. Under the FAIR hypothesis, given $Y = 1$ and $A = a$, $f \sim \text{Ber}(\frac{1}{2})$ and given $Y = 0$ and $A = a$, $f \sim \text{Ber}(\frac{p}{2(1-q)})$ for both $a \in \{0, 1\}$. On the other hand, in the UNFAIR hypothesis, the distribution of $f$ is as specified in Table 1.

| Given | Distribution of $f$ |
|---|---|
| $Y = 1, A = 0$ | $f \sim \text{Ber}\left(\frac{1}{2}\right)$ |
| $Y = 1, A = 1$ | $f \sim \text{Ber}\left(\frac{1}{2(1+4\varepsilon)}\right)$ |
| $Y = 0, A = 0$ | $f \sim \text{Ber}\left(\frac{p}{2(1-q)}\right)$ |
| $Y = 0, A = 1$ | $f \sim \text{Ber}\left(\frac{p}{2(1-q-4\varepsilon q)}\right)$ |

Table 1: Distribution of $f$ under UNFAIR hypothesis.

Using the description of $f$ above, we can generate Table 2 that shows the values of various quantities of interest.

| **Probability** | FAIR (same for both $a$) | UNFAIR | |
|---|---|---|---|
| | | $a = 0$ | $a = 1$ |
| $\mathbb{P}[f = 1, Y = 1 \mid A = a]$ | $\frac{q}{2}$ | $\frac{q}{2}$ | $\frac{q}{2}$ |
| $\mathbb{P}[f = 1, Y = 0 \mid A = a]$ | $\frac{p}{2}$ | $\frac{p}{2}$ | $\frac{p}{2}$ |
| $\mathbb{P}[f = 0, Y = 1 \mid A = a]$ | $\frac{q}{2}$ | $\frac{q}{2}$ | $\frac{q}{2}(1 + 8\varepsilon)$ |
| $\mathbb{P}[f = 0, Y = 0 \mid A = a]$ | $1 - q - \frac{p}{2}$ | $1 - q - \frac{p}{2}$ | $1 - q - \frac{p}{2} - 4\varepsilon q$ |

Table 2: Probability values of interest under the constructed $f$.

From Table 2, it can be seen that given $A = a$, $\mathbb{P}_{\text{FAIR}}[f = 1 \mid A = a] = \mathbb{P}_{\text{UNFAIR}}[f = 1 \mid A = a]$. Moreover, the $\mathbb{P}_{\text{FAIR}}[Y = 1 \mid f = 1, A = a] = \mathbb{P}_{\text{UNFAIR}}[Y = 1 \mid f = 1, A = a]$. Hence, $D_{KL}(\mathbb{P}_{\text{FAIR}}[Y \mid f = 1, A = a] \parallel \mathbb{P}_{\text{UNFAIR}}[Y \mid f = 1, A = a]) = 0$. Finally, $\mathbb{P}_{\text{FAIR}}[Y = 1 \mid f = 0, A = 1] = \frac{q}{2(1-\frac{p+q}{2})}$ and $\mathbb{P}_{\text{UNFAIR}}[Y = 1 \mid f = 0, A = 1] = \frac{q(1+8\varepsilon)}{2(1-\frac{p+q}{2})}$.

We set $\varepsilon \in (0, \frac{1}{4})$, $p \in (0, \frac{1}{2})$ and $q \in (0, \frac{1}{2})$ so that $q(1 + 4\varepsilon) \leq 1$. It is easy to verify that the equalized odds difference under the FAIR hypothesis is 0 (intuitively, the probabilities of the classifier is the same for both the groups $a \in \{0, 1\}$). The equalized

odds disparity gap of the UNFAIR hypothesis is more than $\varepsilon$ as can be seen below.

$$\mathbb{P}[f = 1 \mid Y = 1, A = 0] - \mathbb{P}[f = 1 \mid Y = 1, A = 1]$$
$$= \frac{\mathbb{P}[f = 1, Y = 1 \mid A = 0]}{\mathbb{P}[Y = 1 \mid A = 0]} - \frac{\mathbb{P}[f = 1, Y = 1 \mid A = 1]}{\mathbb{P}[Y = 1 \mid A = 1]}$$
$$= \frac{q/2}{q} - \frac{q/2}{q(1 + 4\varepsilon)}$$
$$= \frac{1}{2}\left(1 - \frac{1}{1 + 4\varepsilon}\right) = \frac{2\varepsilon}{1 + 4\varepsilon} > \varepsilon \qquad\qquad (\varepsilon < \tfrac{1}{4})$$

Next we lower bound the number of samples required by any algorithm to be able to conclude which of the hypothesis is true. Suppose the auditing algorithm $\pi$ is informed that the classifier is biased only for $a = 1$ group. Then, it makes sense for the algorithm to sample only from the distribution conditioned on $A = 1$.

Consider the canonical partial feedback model with hypotheses $M = $ FAIR and $N = $ UNFAIR when the distribution is conditioned on $A = 1$.

Let $\pi$ interact with the model till some stopping time $\tau$ such that $\mathbb{E}_{\mathsf{FAIR},\pi}[\tau] < \infty$ and $\mathbb{E}_{\mathsf{UNFAIR},\pi}[\tau] < \infty$. Here, $\mathbb{P}_{\mathsf{FAIR},\pi}$ and $\mathbb{P}_{\mathsf{UNFAIR},\pi}$ denote distribution of outcome generated by interaction of $\pi$ with the distributions $\mathbb{P}_{\mathsf{FAIR}}$ and $\mathbb{P}_{\mathsf{UNFAIR}}$ respectively. Let ALG $\in \{$FAIR, UNFAIR$\}$ be the output of the algorithm. Moreover, let $\mathbb{P}_{\mathsf{FAIR},\pi}[\mathsf{ALG} = \mathsf{UNFAIR}] < \delta$ and $\mathbb{P}_{\mathsf{UNFAIR},\pi}[\mathsf{ALG} = \mathsf{FAIR}] < \delta$, $i.e.$, the algorithm $\pi$ succeeds in the $(\varepsilon, \delta)$-fairness audit. Then, we have:

$$2\delta > \mathbb{P}_{\mathsf{FAIR},\pi}[\mathsf{ALG} = \mathsf{UNFAIR}] + \mathbb{P}_{\mathsf{UNFAIR},\pi}[\mathsf{ALG} = \mathsf{FAIR}]$$
$$\geq \frac{1}{2}\exp\left(-D_{KL}(\mathbb{P}_{\mathsf{FAIR},\pi}[\mathsf{ALG}] \parallel \mathbb{P}_{\mathsf{UNFAIR},\pi}[\mathsf{ALG}])\right) \qquad\qquad \text{(Lemma 11)}$$
$$\geq \frac{1}{2}\exp\left(-\mathbb{E}_{\mathsf{FAIR}}[L_\tau] \cdot D_{KL}\left(\mathbb{P}_{\mathsf{FAIR}}[Y \mid f = 0, A = 1] \parallel \mathbb{P}_{\mathsf{UNFAIR}}[Y \mid f = 0, A = 1]\right)\right)$$
$$\qquad\qquad \text{(Lemma 8, } D_{KL}(\mathbb{P}_{\mathsf{FAIR}}[Y \mid f = 1, A = a] \parallel \mathbb{P}_{\mathsf{UNFAIR}}[Y \mid f = 1, A = a]) = 0)$$
$$= \frac{1}{2}\exp\left(-\mathbb{E}_{\mathsf{FAIR}}[L_\tau] \cdot D_{KL}\left(\mathrm{Ber}\left(\frac{q}{2(1 - \frac{p+q}{2})}\right) \parallel \mathrm{Ber}\left(\frac{q(1 + 8\varepsilon)}{2(1 - \frac{p+q}{2})}\right)\right)\right)$$
$$\geq \frac{1}{2}\exp\left(-64 \cdot \mathbb{E}_{\mathsf{FAIR}}[L_\tau] \cdot \varepsilon^2 \cdot \frac{q}{2(1 - \frac{p+q}{2})}\right) \quad \text{(Lemma 12, } \frac{q}{2-p-q} \leq \tfrac{1}{2} \text{ and } \frac{8q\varepsilon}{2-p-q} \leq 1 \text{ for } q \in (0, \tfrac{1}{2}) \text{ and } \varepsilon \in (0, \tfrac{1}{4}))$$

Therefore, $\mathbb{E}_{\mathsf{FAIR}}[L_\tau] > \frac{(1 - \frac{p+q}{2})\log(1/4\delta)}{32q\varepsilon^2}$. Moreover, since Lemma 11 is symmetric, we also have $\mathbb{E}_{\mathsf{UNFAIR}}[L_\tau] > \frac{(1 - \frac{p+q}{2})\log(1/4\delta)}{64q\varepsilon^2}$ using Lemma 12. Hence, $\min\{\mathbb{E}_{\mathsf{FAIR}}[L_\tau], \mathbb{E}_{\mathsf{UNFAIR}}[L_\tau]\} > \frac{(1 - \frac{p+q}{2})\log(1/4\delta)}{64q\varepsilon^2}$. Therefore, under both the hypothesis, the algorithm must request $\Omega\left(\frac{\mathbb{P}[f=0|A=1]\log(1/\delta)}{\mathbb{P}[Y=1|A=1]\varepsilon^2}\right)$ samples in expectation to be able to output the correct hypothesis.

Repeating this argument over all combinations of $y \in \{0, 1\}$ and $a \in \mathcal{A}$ concludes the theorem. $\qquad\square$

### A.4   Auxiliary Lemmas

**Lemma 10.** *Suppose $\varepsilon \in (0, \tfrac{1}{2})$. Then we have, (i) $\frac{1-\varepsilon}{1+\varepsilon} \geq 1 - 2\varepsilon$ and (ii) $\frac{1+\varepsilon}{1-\varepsilon} \leq 1 + 4\varepsilon$ .*

*Proof.* For (i), we have

$$\frac{1-\varepsilon}{1+\varepsilon} \geq 1 - 2\varepsilon$$

$$\Longleftrightarrow \quad 1 - \varepsilon \geq (1 - 2\varepsilon)(1 + \varepsilon)$$

$$\Longleftrightarrow \quad (1 - 2\varepsilon)(1 + \varepsilon) = 1 + \varepsilon - 2\varepsilon - 2\varepsilon^2$$

$$\Longleftrightarrow \quad 1 + \varepsilon - 2\varepsilon - 2\varepsilon^2 = 1 - \varepsilon - 2\varepsilon^2$$

$$\Longleftrightarrow \quad 1 - \varepsilon \geq 1 - \varepsilon - 2\varepsilon^2$$

$$\Longleftrightarrow \quad 0 \geq -2\varepsilon^2 \,.$$

This is always true for $\varepsilon \in (0, \frac{1}{2})$, so the inequality holds.

Similarly, for (ii) we have

$$\frac{1+\varepsilon}{1-\varepsilon} \leq 1 + 4\varepsilon$$

$$\Longleftrightarrow \quad 1 + \varepsilon \leq (1 + 4\varepsilon)(1 - \varepsilon)$$

$$\Longleftrightarrow \quad 1 + \varepsilon \leq 1 + 3\varepsilon - 4\varepsilon^2$$

$$\Longleftrightarrow \quad 2\varepsilon(1 - 2\varepsilon) \geq 0 \,.$$

This is clearly true for $\varepsilon \in (0, \frac{1}{2})$, so the inequality holds. $\qquad\square$

**Lemma 11** (Bretagnolle-Huber Inequality (Theorem 14.2 [LS20]))**.** *Let $P$ and $Q$ be two probability measures on some probability space $(\Omega, \mathcal{F})$. Let $A$ be a measurable event in this probability space. Then,*

$$P(A) + Q(A^c) \geq \frac{1}{2}\exp(-D_{KL}(P \parallel Q)) \,.$$

**Lemma 12.** *Let $q \in (0, \frac{1}{2})$ and $\varepsilon \in (0, 1)$ such that $\frac{8q\varepsilon}{1-q} \leq 1$. Then, we have*

$$D_{KL}(\mathrm{Ber}(q) \parallel \mathrm{Ber}(q(1 + 8\varepsilon))) \leq 64q\varepsilon^2 \qquad and \qquad D_{KL}(\mathrm{Ber}(q(1 + 4\varepsilon)) \parallel \mathrm{Ber}(q)) \leq 128q\varepsilon^2 \,.$$

*Proof.*

$$
\begin{aligned}
D_{KL}(\mathrm{Ber}(q) \parallel \mathrm{Ber}(q(1 + 8\varepsilon))) &= q\log\left(\frac{q}{q(1 + 8\varepsilon)}\right) + (1 - q)\log\left(\frac{1 - q}{1 - q(1 + 8\varepsilon)}\right) \\
&= -q\log(1 + 8\varepsilon) - (1 - q)\log\left(1 - \frac{8q\varepsilon}{1 - q}\right) \\
&\leq -q\left(8\varepsilon - \frac{(8\varepsilon)^2}{2}\right) - (1 - q)\left(-\frac{8q\varepsilon}{1 - q} - \frac{(8q\varepsilon)^2}{2(1 - q)^2}\right) \\
&\qquad\qquad (\log(1 + x) \geq x - \tfrac{x^2}{2} \text{ and } \log(1 - x) \geq -x - \tfrac{x^2}{2} \text{ for } x \geq 0) \\
&= \frac{q(8\varepsilon)^2}{2}\left(\frac{q}{1 - q} + 1\right) \\
&= \frac{32q\varepsilon^2}{1 - q} \\
&\leq 64q\varepsilon^2 \qquad\qquad\qquad\qquad\qquad\qquad\qquad\qquad\qquad (q \in (0, \tfrac{1}{2}))
\end{aligned}
$$

For the other inequality, we have,

$$
\begin{aligned}
D_{KL}(\mathrm{Ber}(q(1+8\varepsilon)) \parallel \mathrm{Ber}(q)) &= q(1+8\varepsilon)\log\left(\frac{q(1+8\varepsilon)}{q}\right) + (1-q(1+8\varepsilon))\log\left(\frac{1-q(1+8\varepsilon)}{1-q}\right) \\
&= q(1+8\varepsilon)\log(1+8\varepsilon) + (1-q(1+8\varepsilon))\log\left(1-\frac{8q\varepsilon}{1-q}\right) \\
&\leq q(1+8\varepsilon)(8\varepsilon) + (1-q(1+8\varepsilon))\left(-\frac{8q\varepsilon}{1-q}\right) \qquad (\log(1+x)\leq x \text{ for } x \in (-1,1)) \\
&= 8q\varepsilon + 64q\varepsilon^2 - 8q\varepsilon + \frac{64q^2\varepsilon^2}{1-q} \\
&= 64q\varepsilon^2\left(1+\frac{q}{1-q}\right) \\
&\leq 128q\varepsilon^2 \qquad\qquad (q \in (0, \tfrac{1}{2}))
\end{aligned}
$$

$\square$

# B  Appendix for Mixture Model

Recall that in the mixture model, given $A = a$, the true label is sampled as $Y \sim Ber(q_{1|a})$ and then the feature $X$ is sampled as $X \sim \mathcal{E}_{\theta^*_{y,a}}$, where $\mathcal{E}_{\theta^*_{y,a}}(X) = h(X)\exp(\theta^*_{y,a}T(X) - W(\theta))$. We denote by $\mathcal{B}(a,b)$ the $d$-dimensional ball of radius $b$ around a point $a \in \mathbb{R}^d$. In the mixture model setting, we make the following assumptions: (i) $\mathbb{P}[f = 1 \mid Y = y, A = a] \geq \alpha > 0$ for some known constant $\alpha$, (ii) $\mathcal{E}_\theta(X)$ is log-concave in $X$, $\kappa \mathbf{I} \preceq \nabla^2 W(\theta) \preceq \lambda \mathbf{I}$ with $0 < \kappa < \lambda$ and $W$ is $L$-Lipschitz for all $\theta \in \Theta$, $T(X)$ has components polynomial in $X$, $\mathcal{B}(\theta^*_{y,a}, \frac{1}{\beta}) \subset \Theta$ for some $\beta > 0$ for all $y \in \{0,1\}$ and $a \in \mathcal{A}$. Without loss of generality, let $\kappa \leq 1$ and $L, \beta \geq 1$. We now give the proof of Theorem 4.

## B.1  Proof of Theorem 4

**Theorem 13** (Restatement of Theorem 4). *Given $\varepsilon, \delta \in (0,1)$, under the mixture model, $\mathtt{Exp\text{-}Audit}$ (Algorithm 2) successfully performs the $(\varepsilon, \delta)$-fairness audit with the following cost complexity:*

$$
\mathtt{cost} \leq \tilde{O}\left(\sum_{a \in \mathcal{A}}\left(\frac{c_{feat}\beta_a}{q_{m,a}\varepsilon^2} + \sum_{y \in \{0,1\}}\frac{c_{lab}(q_{0|a} - p_{0|a})}{q_{y|a}}\right)\right).
$$

*Proof.* Firstly, for $\widehat{\theta}_{y,a} = \mathtt{TruncEst}(D_{y,a}, f, \frac{\varepsilon}{2\beta}, \frac{\delta}{14|\mathcal{A}|})$, we have $\|\widehat{\theta}_{y,a} - \theta^*_{y,a}\| \leq \frac{\varepsilon}{2\beta}$ with probability at least $1 - \frac{\delta}{14|\mathcal{A}|}$ due to the guarantees of the $\mathtt{TruncEst}$ subroutine as stated in Theorem 15 and Remark 5. Therefore, the event $\xi_1 = \{\forall\, y \in \{0,1\}, a \in \mathcal{A}, \|\widehat{\theta}_{y,a} - \theta^*_{y,a}\| \leq \frac{\varepsilon}{2\beta}\}$ happens with probability at least $1 - \frac{2\delta}{14}$ via Union Bound.

Next, we have by arguments essentially similar to proof of Theorem 2, the following event $\xi_2$ holds with probability at least $1 - \frac{8\delta}{14}$: for all $y \in \{0,1\}$ and $a \in \mathcal{A}$, $(1 - \frac{\varepsilon}{12})p_{y|a} \leq \widehat{p}_{y|a} \leq (1+\frac{\varepsilon}{12})p_{y|a}$ and $(1-\varepsilon')q_{y|a} \leq \widetilde{q}_{y|a} \leq (1+\varepsilon')q_{y|a}$.

Therefore, $\xi_1 \cap \xi_2$ happens with probability at least $1 - \frac{10\delta}{14}$. Hereon, we will assume that $\xi_1 \cap \xi_2$ holds.

Let us now analyze the MAP oracle for a fixed group label $A = a$. Under event $\xi_1 \cap \xi_2$, we have $\|\widehat{\theta}_{1,a} - \widehat{\theta}_{0,a}\| \geq \|\theta^*_{1,a} - \theta^*_{0,a}\| - 2 \cdot \frac{\varepsilon}{2\beta}$ by triangle inequality. Therefore, by separation condition (1), we have,

$$
\|\widehat{\theta}_{1,a} - \widehat{\theta}_{0,a}\| \geq \sqrt{\frac{3(16(L \vee \beta)\beta + \lambda)}{4\kappa\beta^2}\log\left(\frac{10q_{M,a}}{q_{m,a}\varepsilon}\right)} - \frac{\varepsilon}{\beta} \geq \sqrt{\frac{3(16(L \vee \beta)\beta + \lambda)}{8\kappa\beta^2}\log\left(\frac{10q_{M,a}}{q_{m,a}\varepsilon}\right)},
$$

where we have used the fact that $L, \beta \geq 1$ and $\kappa \leq 1$ and then applied Lemma 16 (1). Hence under $\xi_1 \cap \xi_2$, we have the following for all $a \in \mathcal{A}$:

1. $\|\widehat{\theta}_{y,a} - \theta^*_{y,a}\| \leq \frac{\varepsilon}{2\beta}$ for $y \in \{0, 1\}$,

2. $\|\widehat{\theta}_{1,a} - \widehat{\theta}_{0,a}\| \geq \sqrt{\frac{3(16(L \vee \beta)\beta + \lambda)}{16\kappa\beta^2} \log\left(\frac{10q_{M,a}}{q_{m,a}\varepsilon}\right)}$ , and

3. $\mathcal{B}\left(\frac{\widehat{\theta}_{0,a} + \widehat{\theta}_{1,a}}{2}, \frac{1}{2\beta}\right) \subset \Theta$ by Lemma 19.

Hence, we can now apply Lemma 14 with $\theta^* = \theta^*_{y,a}$, $\theta_1 = \widehat{\theta}_{y,a}$, $\theta_2 = \widehat{\theta}_{1-y,a}$, $\gamma = \frac{10q_{M,a}}{q_{m,a}} \geq e^2$ and $t = \frac{\widetilde{q}_{1-y|a}}{\widetilde{q}_{y|a}}$. Note that all the conditions on $W$ are satisfied by assumption. Suppose $X \sim \mathcal{E}_{\theta^*_{y,a}}$. Then, we have the following:

$$
\begin{aligned}
\mathbb{P}_X\left[\widetilde{q}_{y|a}\mathcal{E}_{\widehat{\theta}_{y,a}}(X) < \widetilde{q}_{1-y|a}\mathcal{E}_{\widehat{\theta}_{1-y,a}}(X)\right] &= \mathbb{P}_X\left[\sqrt{\frac{\mathcal{E}_{\widehat{\theta}_{1-y,a}}(X)}{\mathcal{E}_{\widehat{\theta}_{y,a}}(X)}} > \sqrt{\frac{\widetilde{q}_{y|a}}{\widetilde{q}_{1-y|a}}}\right] \\
&\leq \mathbb{P}_X\left[\sqrt{\frac{\mathcal{E}_{\widehat{\theta}_{1-y,a}}(X)}{\mathcal{E}_{\widehat{\theta}_{y,a}}(X)}} > \sqrt{\frac{(1-\varepsilon')q_{y|a}}{(1+\varepsilon')q_{1-y|a}}}\right] \\
&\qquad\text{(Event } \xi_1 \cap \xi_2 \text{ holds, } \widetilde{q}_{y|a} \in (1 \pm \varepsilon')q_{y|a} \ \forall \ y \in \{0, 1\}, a \in \mathcal{A}) \\
&\leq \sqrt{\frac{(1-\varepsilon')q_{y|a}}{(1+\varepsilon')q_{1-y|a}} \cdot \left(\frac{\varepsilon q_{m,a}}{10q_{M,a}}\right)^2} \qquad\qquad \text{(by Lemma 14)} \\
&\leq \frac{\varepsilon q_{m,a}}{36} \ . \qquad\qquad\qquad\qquad\qquad\qquad \text{(Plugging } \varepsilon' = \frac{1}{10})
\end{aligned}
$$

Therefore, the approximate MAP oracle $M_a(X) = \arg\max_{y \in \{0,1\}} \widetilde{q}_{y|a}\mathcal{E}_{\widehat{\theta}_{y|a}}(X)$ correctly classifies at least $1 - \frac{\varepsilon q_{m,a}}{36}$ fraction of the samples generated from $\mathcal{E}_{\theta^*_{y|a}}$ for every $y \in \{0, 1\}$. For a given $y$, recall that $R_y$ (line 10 of Algorithm 2) counts the number of samples assigned label $y$ by the approximate MAP oracle. For $X_1, X_2, \ldots, X_R$ drawn iid according to the mixture model $\mathcal{M}_a := \sum_{y \in \{0,1\}} q_{y|a}\mathcal{E}_{\theta^*_{y,a}}$, we have

$$\mathbb{E}[R_y] = \mathbb{E}\left[\sum_{i=1}^{R} \mathbb{1}\{M_a(X_i) = y\}\right]$$

$$= \sum_{i=1}^{R} \mathbb{E}_{X_i \sim \mathcal{M}_a}[\mathbb{1}\{M_a(X_i) = y\}]$$

$$= \sum_{i=1}^{R} \mathbb{P}_{X \sim \mathcal{M}_a}[M_a(X) = y] \qquad\qquad\qquad (X_i\text{'s are iid})$$

$$= R \cdot \mathbb{P}_{X \sim \mathcal{M}_a}[M_a(X) = y]$$

$$= R \cdot \left(\mathbb{P}_{X \sim \mathcal{M}_a}[M_a(X) = y \mid Y = y] \cdot \mathbb{P}[Y = y] + \mathbb{P}_{X \sim \mathcal{M}_a}[M_a(X) = y \mid Y = 1 - y] \cdot \mathbb{P}[Y = 1 - y]\right)$$

$$= R \cdot \left(\mathbb{P}_{X \sim \mathcal{E}_{\theta_{y,a}^*}}[M_a(X) = y] \cdot \mathbb{P}[Y = y] + \mathbb{P}_{X \sim \mathcal{E}_{\theta_{1-y,a}^*}}[M_a(X) = y] \cdot \mathbb{P}[Y = 1 - y]\right)$$

$$\geq R \cdot \mathbb{P}_{X \sim \mathcal{E}_{\theta_{y,a}^*}}[M_a(X) = y] \cdot \mathbb{P}[Y = y]$$

$$\geq R \cdot (1 - \frac{\varepsilon q_{m,a}}{36})q_{y|a} \geq R \cdot (1 - \frac{\varepsilon}{36})q_{y|a}$$

Similarly, we also have,

$$\mathbb{E}[R_y] = R \cdot \left(\mathbb{P}_{X \sim \mathcal{E}_{\theta_{y,a}^*}}[M_a(X) = y] \cdot \mathbb{P}[Y = y] + \mathbb{P}_{X \sim \mathcal{E}_{\theta_{1-y,a}^*}}[M_a(X) = y] \cdot \mathbb{P}[Y = 1 - y]\right)$$

$$\leq R \cdot \left(1 \cdot q_{y|a} + \frac{\varepsilon q_{m,a}}{36} \cdot q_{1-y|a}\right) \leq R \cdot (1 + \frac{\varepsilon}{36})q_{y|a}$$

Therefore, via an application of the Chernoff Bound (Lemma 28),

$$\mathbb{P}\left[(1 - \varepsilon_o)(1 - \frac{\varepsilon}{36})q_{y|a}R \leq R_y \leq (1 + \varepsilon_o)(1 + \frac{\varepsilon}{36})q_{y|a}R\right] \geq 1 - 2\exp\left(-\frac{\varepsilon_o^2 \cdot \mathbb{E}[R_y]}{3}\right)$$

$$\geq 1 - 2\exp\left(-\frac{\varepsilon_o^2(1 - \frac{\varepsilon}{36})R \cdot q_{y|a}}{3}\right) \qquad (7)$$

In Exp-Audit (Algorithm 2), we have set $R = \frac{3430\log(14|\mathcal{A}|/\delta)}{\min\{\widetilde{q}_{1|a}, \widetilde{q}_{0|a}\}\varepsilon^2}$. Note that under event $\xi_1 \cap \xi_2$, $\widetilde{q}_{y|a} \leq 1.1 q_{y|a}$. Therefore, $\min\{\widetilde{q}_{1|a}, \widetilde{q}_{0|a}\} \leq 1.1 \min\{q_{1|a}, q_{0|a}\} = 1.1 q_{m,a}$. Therefore, $R \geq \frac{3\log(14|\mathcal{A}|/\delta)}{q_{m,a}(1 - \frac{\varepsilon}{36})\left(\frac{\varepsilon}{18.5}\right)^2}$, since $\varepsilon \in (0, 1)$. This implies that in Eq. (7), if we substitute $\varepsilon_o = \frac{\varepsilon}{18.5}$, we have,

$$\mathbb{P}\left[(1 - \frac{\varepsilon}{18.5})(1 - \frac{\varepsilon}{36})q_{y|a}R \leq R_y \leq (1 + \frac{\varepsilon}{18.5})(1 + \frac{\varepsilon}{36})q_{y|a}R\right] \geq 1 - \frac{2\delta}{14|\mathcal{A}|} .$$

Hence, the estimate $\widehat{q}_{y|a}$ obtained in Line 11 of Exp-Audit (Algorithm 2) for all $a \in \mathcal{A}$ satisfies with probability at least $1 - \frac{4\delta}{14}$ (Union Bound): $(1 - \frac{\varepsilon}{12})q_{y|a} \leq \widehat{q}_{y|a} \leq (1 + \frac{\varepsilon}{12})q_{y|a}$ for all $y \in \{0, 1\}$, where we have used $1 - \frac{\varepsilon}{12} \leq (1 - \frac{\varepsilon}{36})(1 - \frac{\varepsilon}{18.5})$ and $1 + \frac{\varepsilon}{12} \geq (1 + \frac{\varepsilon}{36})(1 + \frac{\varepsilon}{18.5})$ for all $\varepsilon \in (0, 1)$.

Having found an estimate $\widehat{q}_{y|a} \in (1 \pm \frac{\varepsilon}{12})q_{y|a}$, the guarantee in terms of the correctness of the hypothesis test follows similarly as in the proof of Theorem 2. Hence, the algorithm is successful in the $(\varepsilon, \delta)$-fairness audit is with probability at least $1 - \delta$.

The audit cost can be calculated by observing that the audit cost is incurred in two steps of the algorithm—first to obtain $\widetilde{q}_{y|a}$ and then to observe $R$ features. For estimating $\widetilde{q}_{y|a}$, the auditor pays $\tilde{O}\left(\frac{c_{feat}\beta_a + c_{lab}(q_{0|a} - p_{0|a})}{q_{y|a}\varepsilon'^2}\right)$ (similar to proof of Theorem 2).

Since $\varepsilon' = \frac{1}{10}$ is a constant, the cost of estimating $\widetilde{q}_{y|a}$ is $\widetilde{O}\left(\frac{c_{feat}\beta_a + c_{lab}(q_{0|a} - p_{0|a})}{q_{y|a}}\right)$. Adding this cost for both $y = 0$ and $y = 1$ gives that the audit cost of this stage of the algorithm for a particular group $A = a$ is $\widetilde{O}\left(\sum_{y \in \{0,1\}} \frac{c_{feat}\beta_a + c_{lab}(q_{0|a} - p_{0|a})}{q_{y|a}}\right)$.

For observing $R$ features, the auditor incurs an audit cost of $R \cdot c_{feat} \cdot \beta_a$. Plugging $R = \widetilde{O}\left(\frac{1}{q_{m,a}\varepsilon^2}\right)$, we have that the audit cost for this phase is $\widetilde{O}\left(\frac{c_{feat}\beta_a}{q_{m,a}\varepsilon^2}\right)$.

Summing the two costs up for all group labels $a \in \mathcal{A}$ we have the desired result. $\qquad\square$

**Lemma 14.** *Let $\Theta \subset \mathbb{R}^d$ be a convex set containing parameters $\theta_1$, $\theta_2$ and $\theta^*$. Let $\varepsilon \in (0,1)$ and $L, \beta \geq 1$ and $\kappa \leq 1$. Further, assume that (i) $\kappa \mathbf{I} \preceq \nabla^2 W(\theta) \preceq \lambda \mathbf{I}$, (ii) $W(\theta)$ is L-Lipschitz for all $\theta \in \Theta$, (iii) $\|\theta_1 - \theta_2\| \geq \sqrt{\frac{3(16(L \vee \beta)\beta + \lambda)}{16\kappa\beta^2} \log\left(\frac{\gamma}{\varepsilon}\right)}$ for some $\gamma \geq e^2$, (iv) $\|\theta_1 - \theta^*\| \leq \frac{\varepsilon}{2\beta}$, and (v) $\mathcal{B}\left(\frac{\theta_1 + \theta_2}{2}, \frac{1}{2\beta}\right) \subset \Theta$. Then we have $\mathbb{P}_{X \sim \mathcal{E}_{\theta^*}}[\mathcal{E}_{\theta_1}(X) > t \cdot \mathcal{E}_{\theta_2}(X)] \geq 1 - \sqrt{t}\left(\frac{\varepsilon}{\gamma}\right)^2$.*

*Proof.* We have the following:

$$
\mathbb{P}_{X \sim \mathcal{E}_{\theta^*}}[\mathcal{E}_{\theta_1}(X) < t \cdot \mathcal{E}_{\theta_2}(X)]
$$
$$
= \mathbb{P}_{X \sim \mathcal{E}_{\theta^*}}\left[\sqrt{\frac{\mathcal{E}_{\theta_2}(X)}{\mathcal{E}_{\theta_1}(X)}} > \sqrt{\frac{1}{t}}\right]
$$
$$
\leq \sqrt{t} \cdot \mathbb{E}_{X \sim \mathcal{E}_{\theta^*}}\left[\sqrt{\frac{\mathcal{E}_{\theta_2}(X)}{\mathcal{E}_{\theta_1}(X)}}\right] \qquad\qquad\qquad\text{(Markov's Inequality)}
$$
$$
= \sqrt{t} \int_{x \in \mathcal{X}} h(x) \exp(\theta^{*\intercal}T(x) - W(\theta^*)) \cdot \sqrt{\frac{h(x)\exp(\theta_2^\intercal T(x) - W(\theta_2))}{h(x)\exp(\theta_1^\intercal T(x) - W(\theta_1))}} \cdot dx
$$
$$
= \sqrt{t} \int_{x \in \mathcal{X}} h(x) \exp(\theta^{*\intercal}T(x) - W(\theta^*)) \cdot \exp\left(\frac{1}{2}\left((\theta_2 - \theta_1)^\intercal T(x) - W(\theta_2) + W(\theta_1)\right)\right) \cdot dx
$$
$$
= \sqrt{t} \int_{x \in \mathcal{X}} h(x) \exp\left(\left(\frac{\theta_1 + \theta_2}{2}\right)^\intercal T(x) + (\theta^* - \theta_1)^\intercal T(x) - \frac{W(\theta_1) + W(\theta_2)}{2} + W(\theta_1) - W(\theta^*)\right) \cdot dx
$$

We now explain the strategy to bound the above quantity. Firstly, we will use the Lipschitzness of $W$ to bound $W(\theta_1) - W(\theta^*)$ by leveraging the fact that $\theta_1$ and $\theta^*$ are close by assumption of the lemma. Next, the goal is to bound the remaining expressions so that we obtain a valid probability density function from the exponential family, possibly with a different parameter than what we started with. At this point, one may be tempted to use the convexity of $W$ to lower bound $\frac{W(\theta_1) + W(\theta_2)}{2} \geq W\left(\frac{\theta_1 + \theta_2}{2}\right)$. Although this give us the probability density whose parameter is $\frac{\theta_1 + \theta_2}{2}$, the bound is not tight and does not allow us to obtain

meaningful results. Therefore, we resort to bounding $\frac{W(\theta_1)+W(\theta_2)}{2}$ via Taylor's theorem. The steps are illustrated below.

$$\mathbb{P}_X[\mathcal{E}_{\theta_1}(X) < t \cdot \mathcal{E}_{\theta_2}(X)]$$

$$\leq \sqrt{t} e^{\frac{L\varepsilon}{2\beta}} \int_{x \in \mathcal{X}} h(x) \exp\left(\left(\frac{\theta_1+\theta_2}{2}\right)^{\mathsf{T}} T(x) - \frac{W(\theta_1)+W(\theta_2)}{2}\right) \cdot \exp\left((\theta^*-\theta_1)^{\mathsf{T}} T(x)\right) \cdot dx$$

$$(\|\theta^*-\theta_1\| \leq \tfrac{\varepsilon}{2\beta}, W \text{ is } L\text{-Lipschitz})$$

$$= \sqrt{t} e^{\frac{L\varepsilon}{2\beta}} \int_{x \in \mathcal{X}} h(x) \exp\left(\left(\frac{\theta_1+\theta_2}{2}\right)^{\mathsf{T}} T(x) - W\left(\frac{\theta_1+\theta_2}{2}\right) - \frac{1}{4}\|\theta_1-\theta_2\|^2_{(\nabla^2 W(\bar{\theta})+\nabla^2 W(\tilde{\theta}))}\right) \exp\left((\theta^*-\theta_1)^{\mathsf{T}} T(x)\right) dx$$

$$(\text{Lemma 17, } \bar{\theta} \in [\theta_1, \tfrac{\theta_1+\theta_2}{2}], \tilde{\theta} \in [\theta_2, \tfrac{\theta_1+\theta_2}{2}])$$

$$= \sqrt{t} \exp\left(\frac{L\varepsilon}{2\beta} - \frac{1}{4}\|\theta_1-\theta_2\|^2_{(\nabla^2 W(\bar{\theta})+\nabla^2 W(\tilde{\theta}))}\right)$$

$$\cdot \int_{x \in \mathcal{X}} h(x) \exp\left(\left(\frac{\theta_1+\theta_2}{2}\right)^{\mathsf{T}} T(x) - W\left(\frac{\theta_1+\theta_2}{2}\right)\right) \cdot \exp\left((\theta^*-\theta_1)^{\mathsf{T}} T(x)\right) \cdot dx$$

After this, we will use $\nabla^2 W(\theta) \succeq \kappa \mathbf{I}$ for all $\theta \in \Theta$ to lower bound the quadratic term. On the other hand, the quantity in the integral is actually an expectation with respect to the distribution whose parameter is $\frac{\theta_1+\theta_2}{2}$. To bound the $\exp\left((\theta^*-\theta_1)^{\mathsf{T}} T(x)\right)$ in expectation, we will use the sub-exponential property of the exponential family distribution combined with the fact that for the exponential family we have $\mathbb{E}_{X \sim \mathcal{E}_\theta}[T(X)] = \nabla W(\theta)$.

$$\mathbb{P}_X[\mathcal{E}_{\theta_1}(X) < t \cdot \mathcal{E}_{\theta_2}(X)]$$

$$\leq \sqrt{t} \exp\left(\frac{L\varepsilon}{2\beta} - \frac{1}{4}\|\theta_1-\theta_2\|^2_{(\nabla^2 W(\bar{\theta})+\nabla^2 W(\tilde{\theta}))} + (\theta^*-\theta_1)^{\mathsf{T}} \mu\right) \cdot \mathbb{E}_{X \sim \mathcal{E}_{\frac{\theta_1+\theta_2}{2}}}\left[\exp\left((\theta^*-\theta_1)^{\mathsf{T}}(T(X)-\mu)\right]\right.$$

$$(\mu := \mathbb{E}_{X \sim \mathcal{E}_{\frac{\theta_1+\theta_2}{2}}}[T(X)])$$

$$\leq \sqrt{t} \cdot \exp\left(\frac{L\varepsilon}{2\beta} - \frac{1}{4}\|\theta_1-\theta_2\|^2_{2\kappa \mathbf{I}} + (\theta^*-\theta_1)^{\mathsf{T}} \mu\right) \cdot \mathbb{E}_{X \sim \mathcal{E}_{\frac{\theta_1+\theta_2}{2}}}\left[\exp\left((\theta^*-\theta_1)^{\mathsf{T}}(T(X)-\mu)\right]\right.$$

$$(\nabla^2 W(\theta) \succeq \kappa \mathbf{I} \text{ for all } \theta \in \Theta)$$

$$\leq \sqrt{t} \cdot \exp\left(\frac{L\varepsilon}{2\beta} - \frac{\kappa}{2}\|\theta_1-\theta_2\|^2 + \|\theta^*-\theta_1\| \cdot \|\mu\|\right) \cdot \mathbb{E}_{X \sim \mathcal{E}_{\frac{\theta_1+\theta_2}{2}}}\left[\exp\left((\theta^*-\theta_1)^{\mathsf{T}}(T(X)-\mu)\right]\right.$$

$$(\text{Cauchy-Schwarz Inequality})$$

$$\leq \sqrt{t} \cdot \exp\left(\frac{L\varepsilon}{\beta} - \frac{\kappa}{2}\|\theta_1-\theta_2\|^2\right) \cdot \mathbb{E}_{X \sim \mathcal{E}_{\frac{\theta_1+\theta_2}{2}}}\left[\exp\left((\theta^*-\theta_1)^{\mathsf{T}}(T(X)-\mu)\right]\right.$$

$$(\|\theta^*-\theta_1\| \leq \tfrac{\varepsilon}{2\beta}; \|\mu\| = \|\nabla W\left(\tfrac{\theta_1+\theta_2}{2}\right)\| \leq L \text{ since } W \text{ is } L\text{-Lipschitz})$$

$$\leq \sqrt{t} \cdot \exp\left(\frac{L\varepsilon}{\beta} - \frac{\kappa}{2}\|\theta_1-\theta_2\|^2\right) \cdot \exp\left(\frac{\|\theta^*-\theta_1\|^2 \lambda}{2}\right)$$

$$(\text{Lemma 18 as by assumption } \mathcal{B}\left(\tfrac{\theta_1+\theta_2}{2}, \tfrac{1}{2\beta}\right) \subset \Theta \text{ and } \|\theta^*-\theta_1\| < \tfrac{1}{2\beta})$$

$$\leq \sqrt{t} \cdot \exp\left(\frac{(L \vee \beta)\varepsilon}{\beta} + \frac{\lambda\varepsilon^2}{8\beta^2} - \frac{\kappa}{2}\|\theta_1-\theta_2\|^2\right)$$

$$(\|\theta^*-\theta_1\| \leq \tfrac{\varepsilon}{2\beta} \text{ and } L \leq (L \vee \beta))$$

Finally, we will use the assumption in the lemma statement that $\|\theta_1 - \theta_2\| \geq \sqrt{\frac{3(16(L \vee \beta)\beta + \lambda)}{16\kappa\beta^2} \log\left(\frac{\gamma}{\varepsilon}\right)}$. Hence, we obtain

$$
\begin{aligned}
&\mathbb{P}_X[\mathcal{E}_{\theta_1}(X) < t \cdot \mathcal{E}_{\theta_2}(X)] \\
&\leq \sqrt{t} \cdot \exp\left(\frac{(L \vee \beta)\varepsilon}{\beta} + \frac{\lambda\varepsilon^2}{8\beta^2} - \frac{\kappa}{2} \cdot \frac{3(16(L \vee \beta)\beta + \lambda)}{8\kappa\beta^2} \log\left(\frac{\gamma}{\varepsilon}\right)\right) && (\|\theta_1 - \theta_2\| \geq \sqrt{\tfrac{3(16(L\vee\beta)\beta+\lambda)}{8\kappa\beta^2} \log\left(\tfrac{\gamma}{\varepsilon}\right)}) \\
&\leq \sqrt{t} \cdot \exp\left(-\frac{16(L \vee \beta)\beta + \lambda}{8\beta^2}\left(\log\left(\frac{\gamma}{\varepsilon}\right) - \varepsilon\right) - \frac{16(L \vee \beta)\beta + \lambda}{16\beta^2} \log\left(\frac{\gamma}{\varepsilon}\right)\right) && (\varepsilon^2 \leq \varepsilon \text{ for } \varepsilon \in (0,1)) \\
&\leq \sqrt{t} \cdot \exp\left(-\frac{16(L \vee \beta)\beta + \lambda}{16\beta^2} \log\left(\frac{\gamma}{\varepsilon}\right) - \frac{16(L \vee \beta)\beta + \lambda}{16\beta^2} \log\left(\frac{\gamma}{\varepsilon}\right)\right) && (\text{Lemma } 16 \text{ (2)}) \\
&= \sqrt{t} \cdot \left(\frac{\varepsilon}{\gamma}\right)^{\frac{16(L\vee\beta)\beta+\lambda}{8\beta^2}} \leq \sqrt{t} \cdot \left(\frac{\varepsilon}{\gamma}\right)^2
\end{aligned}
$$

$\square$

## B.2 Parameter Estimation from Truncated Samples

In this section we state the `TruncEst` subroutine and its associated guarantees. The subroutine is same as Algorithm 1 of [LWZ23] and its guarantee is a restatement of Theorem 3.1 of [LWZ23].

---

**Algorithm 4** `TruncEst`$(D, f, \varepsilon, \delta, \mathbf{c}, \Theta)$: Procedure to estimate parameters from truncated samples

---

1: Set $\varepsilon \leftarrow \varepsilon/\sqrt{3}$. Set $n$, $m$ and $\eta$ as defined in Theorem 15 (requires $\mathbf{c}$). Initialize $\widehat{\Theta} = \emptyset$.
2: Using $n$ samples from $D$, find the vanilla maximum likelihood estimate of the parameter. Let this estimate be $\theta_0$, that is, $\mathbb{E}_{X \sim \mathcal{E}_{\theta_0}}[T(X)] = \frac{1}{n}\sum_{i=1}^{n} T(X_i)$ where $\{X_i\}_{i=1}^{n}$ are the $n$ samples from $D$.
3: **for** $j = 1, \ldots, 10m\lceil\log(1/\delta)\rceil$ **do**
4:     $Z_j = (n + j)$-th sample from $D$ .
5:     $v_{j-1} = $ `SampleGradient`$(Z_j, \theta_{j-1}, f)$.
6:     $\theta_j = \theta_{j-1} - \eta \cdot v_{j-1}$ .
7:     Project $\theta_j$ to $\{\theta \in \mathbb{R}^d : \|\theta - \theta_0\| \leq \frac{\varepsilon^2 + 2\beta\log(1/\alpha)}{\kappa}\} \cap \Theta$ .
8:     **if** $j \mod m = 0$ **then**
9:         Set $\widehat{\Theta} \leftarrow \widehat{\Theta} \cup \{\theta_j\}$ and $\theta_j \leftarrow \theta_0$.
10: Return $\widehat{\theta} \in \widehat{\Theta}$ such that $|\{\theta \in \widehat{\Theta} : \|\widehat{\theta} - \theta\| \leq 2\varepsilon\sqrt{3}\}| > \frac{1}{2}|\widehat{\Theta}|$ .

---

**Algorithm 5** `SampleGradient`$(Z, \theta, f)$

---

1: **while** True **do**
2:     Sample $Z' \sim \mathcal{E}_\theta$
3:     **if** $f(Z') = 1$ **then**
4:         Return $T(Z') - T(Z)$

---

Let there be a convex set of parameters $\Theta$ such that the data $D$ is generated from an exponential family distribution $\mathcal{E}_{\theta^*}$, $\theta^* \in \Theta$. Further, suppose the following holds:

    1. $\kappa\mathbf{I} \preceq \nabla^2 W(\theta) \preceq \lambda\mathbf{I}$ for all $\theta \in \Theta$.
    2. There exists a finite $\beta$ such that $\mathcal{B}(\theta^*, \frac{1}{\beta}) \in \Theta$.[3] Moreover, wlog assume $\beta \geq 1$.

---

[3]This assumption is implicit in [LWZ23]. The constant in the $O(\cdot)$ depends on $\beta$ in their work.

3. $\mathbb{P}_{X \sim \mathcal{E}_{\theta^*}}[f(X) = 1] \geq \alpha$ for some constant $\alpha > 0$.

4. The probability density function $\mathcal{E}_{\theta^*}(X)$ is log-concave in $X$.

5. $T(X) \in \mathbb{R}^d$ has components which are polynomial in $X$. Further, the maximum degree of such polynomials is at most $k$.

Now we define the following quantities: $d(\alpha) := \varepsilon^2 + 2\beta \log\left(\frac{1}{\alpha}\right)$, $\kappa_f := \frac{1}{2}\left(\frac{\alpha^2 \exp\left(-6\frac{\lambda}{\kappa^2} \cdot d(\alpha)^2\right)}{4Ck}\right)^{2k} \cdot \kappa$, $\lambda_f := \frac{\exp\left(6\frac{\lambda}{\kappa^2} \cdot d(\alpha)^2\right)}{\alpha^2} \cdot \lambda$,

$\rho^2 := d(\lambda_f + \lambda) + \left(1 + \frac{2\lambda}{\kappa}\right)^2 \left(\frac{12\beta\lambda}{\kappa^2} \cdot d(\alpha)^2 - 4\beta\log(\alpha) + \varepsilon^2\right)^2$, $G(d, \lambda, \kappa, \alpha, \beta) := \frac{\rho^2}{\kappa_f^2 \varepsilon^2}$. Here, $C$ is the same constant as described in [LWZ23]. We denote by $\mathbf{c} := (L, \beta, \kappa, \alpha)$.

**Theorem 15** (Theorem 3.1 of [LWZ23])**.** *Under the assumptions stated above, when* $\mathtt{TruncEst}(D, f, \varepsilon, \delta)$ *is run with the values of* $n \geq \frac{2\beta \log(1/\delta)}{\varepsilon^2}$, $\eta = \frac{\kappa_f \varepsilon^2}{2\rho^2} \wedge \frac{1}{\kappa_f}$ *and* $m \geq \left(G(d, \lambda, \kappa, \alpha, \beta) \vee \frac{1}{2}\right) \log\left(\frac{d(\alpha)}{\kappa\varepsilon^2}\right)$, *then we have* $\mathbb{E}\left[\|\theta - \theta^*\|^2\right] \leq \varepsilon^2$ *for all* $\theta \in \widehat{\Theta}$. *Therefore, via Markov's inequality, we also have for any* $\theta \in \widehat{\Theta}$, $\mathbb{P}\left[\|\theta - \theta^*\|^2 \geq 3\varepsilon^2\right] \leq \frac{1}{3}$.

*Remark* 5. By repeating the projected gradient steps (lines 3-8) $O(\log(1/\delta))$ times with fresh data every time, the probability of success can be boosted to $1 - \delta$. Suppose $\widehat{\Theta} = \{\theta_{m,1}, \ldots, \theta_{m,v}\}$, where $v = O(\log(1/\delta))$, be set of candidate solutions. We choose $\widehat{\theta} = \theta_{m,i}$ such that $\|\theta_{m,i} - \theta_{m,j}\| \leq 2\varepsilon\sqrt{3}$ for at least $v/2$ such $j$'s, $j \neq i$ (see Section 3.4.5 of [DGTZ18]). This ensures that with probability at least $1 - \delta$, $\|\widehat{\theta} - \theta^*\| \leq \varepsilon\sqrt{3}$.

### B.3 Some Technical Lemmas

**Lemma 16.** *Suppose* $\varepsilon \in (0, 1]$. *Then we have the following:*

1. $\left(\sqrt{12} - \sqrt{6}\right)\sqrt{a \log\left(\frac{10b}{\varepsilon}\right)} \geq \varepsilon$ *for all* $a, b \geq 1$.

2. $\frac{1}{2}\log\left(\frac{\gamma}{\varepsilon}\right) \geq \varepsilon$ *for all* $\gamma \geq e^2$.

*Proof.* For (1), let $c := \sqrt{12} - \sqrt{6} \geq 1$. Define $f(\varepsilon) := c\sqrt{a \log(10b/\varepsilon)} - \varepsilon$. Then we have $f(1) = c\sqrt{a \log(10b)} - 1 \geq 0$ since $\log(10) \geq 2$ and $a, b, c \geq 1$. Moreover, we have $f'(\varepsilon) = -\frac{c\sqrt{a}}{2\varepsilon\sqrt{\log(10b/\varepsilon)}} - 1 < 0$. Therefore, $f(\varepsilon) \geq f(1) \geq 0$ for all $\varepsilon \in (0, 1]$.

For (2), we observe that for the LHS, $\frac{1}{2}\log(\gamma/\varepsilon) = \frac{1}{2}\log(\gamma) + \frac{1}{2}\log(1/\varepsilon) \geq 1 + \frac{1}{2}\log(1/\varepsilon)$ since $\gamma \geq e^2$. However, for the RHS, $\varepsilon \leq 1$. Hence, the inequality holds. $\qquad\square$

**Lemma 17.** *Let* $W : \Theta \to \mathbb{R}$ *be a convex function defined on a convex set* $\Theta$. *For* $\theta_1, \theta_2 \in \Theta$, *we have*

$$\frac{W(\theta_1) + W(\theta_2)}{2} = W\left(\frac{\theta_1 + \theta_2}{2}\right) + \frac{1}{4}\|\theta_1 - \theta_2\|_{\nabla^2 W(\overline{\theta}) + \nabla^2 W(\widetilde{\theta})}^2,$$

*where* $\overline{\theta}$ *lies on the line joining* $\theta_1$ *and* $\frac{\theta_1 + \theta_2}{2}$, *and* $\widetilde{\theta}$ *lies on the line joining* $\theta_2$ *and* $\frac{\theta_1 + \theta_2}{2}$.

*Proof.* We have the following due to Taylor's Theorem,

$$W(\theta_1) = W\left(\frac{\theta_1 + \theta_2}{2}\right) + \nabla W\left(\frac{\theta_1 + \theta_2}{2}\right)^{\mathsf{T}}\left(\theta_1 - \frac{\theta_1 + \theta_2}{2}\right) + \frac{1}{2}\left\|\theta_1 - \frac{\theta_1 + \theta_2}{2}\right\|_{\nabla^2 W(\overline{\theta})}^2$$

$$= W\left(\frac{\theta_1 + \theta_2}{2}\right) + \nabla W\left(\frac{\theta_1 + \theta_2}{2}\right)^{\mathsf{T}}(\theta_1 - \theta_2) + \frac{1}{2}\|\theta_1 - \theta_2\|_{\nabla^2 W(\overline{\theta})}^2$$

where $\overline{\theta}$ lies on the line joining $\theta_1$ and $\frac{\theta_1 + \theta_2}{2}$. Similarly, we have

$$W(\theta_2) = W\left(\frac{\theta_1 + \theta_2}{2}\right) + \nabla W\left(\frac{\theta_1 + \theta_2}{2}\right)^\top (\theta_2 - \theta_1) + \frac{1}{2}\|\theta_2 - \theta_1\|^2_{\nabla^2 W(\widetilde{\theta})}$$

for some $\widetilde{\theta}$ on the line joining $\theta_2$ and $\frac{\theta_1 + \theta_2}{2}$. Adding the two equations and dividing by 2 gives the desired result. $\qquad\square$

**Lemma 18** (Claim 1 of [LWZ23]). *Let $\theta^*$ lie in a convex set $\Theta \subset \mathbb{R}^d$ and suppose there exists (wlog) $\beta \geq 1$ such that $\mathcal{B}(\theta^*, \frac{1}{\beta}) \subset \Theta$. Let $\mu := \mathbb{E}_{X \sim \mathcal{E}_{\theta^*}}[T(X)]$ and assume that $\nabla^2 W(\theta) \preceq \lambda\mathbf{I}$ for all $\theta \in \Theta$. Then for any vector $\mathbf{u} \in \mathbb{R}^d$ with $\|\mathbf{u}\| < \frac{1}{\beta}$,*

$$\mathbb{E}_{X \sim \mathcal{E}_{\theta^*}}[\exp(\mathbf{u}^\top(T(X) - \mu))] \leq \exp\left(\frac{\|\mathbf{u}\|^2 \lambda}{2}\right) .$$

**Lemma 19.** *Suppose $\theta_1^*$ and $\theta_2^*$ lie in a convex set $\Theta \subset \mathbb{R}^d$ such that $\mathcal{B}(\theta_i^*, \frac{1}{\beta}) \subset \Theta$ for all $i \in \{1, 2\}$. Let $\theta_1, \theta_2 \in \Theta$ be such that $\|\theta_i^* - \theta_i\| \leq \frac{\varepsilon}{2\beta}, \varepsilon \in (0, 1),$ for $i \in \{1, 2\}$. Then, $\mathcal{B}\left(\frac{\theta_1 + \theta_2}{2}, \frac{1}{2\beta}\right) \subset \Theta$.*

*Proof.* Firstly, observe that $\mathcal{B}\left(\frac{\theta_1^* + \theta_2^*}{2}, \frac{1}{\beta}\right) \subset \Theta$ because for any point $\frac{\theta_1^* + \theta_2^*}{2} + r\mathbf{u} \in \mathcal{B}\left(\frac{\theta_1^* + \theta_2^*}{2}, \frac{1}{\beta}\right)$, where $r \in [0, \frac{1}{\beta}]$ and $\mathbf{u}$ is a unit vector, we have,

$$\frac{\theta_1^* + \theta_2^*}{2} + r\mathbf{u} = \frac{1}{2}(\theta_1^* + r\mathbf{u}) + \frac{1}{2}(\theta_2^* + r\mathbf{u}) .$$

Note that $\theta_i^* + r\mathbf{u} \in \mathcal{B}(\theta_i^*, \frac{1}{\beta})$ for $i \in \{1, 2\}$. Hence we have $\theta_i^* + r\mathbf{u} \in \Theta$ for $i \in \{1, 2\}$. Finally by convexity of $\Theta$, we have $\frac{1}{2}(\theta_1^* + r\mathbf{u}) + \frac{1}{2}(\theta_2^* + r\mathbf{u}) \in \Theta$. This implies that $\mathcal{B}\left(\frac{\theta_1^* + \theta_2^*}{2}, \frac{1}{\beta}\right) \subset \Theta$.

Now, observe that $\left\|\left(\frac{\theta_1^* + \theta_2^*}{2}\right) - \left(\frac{\theta_1 + \theta_2}{2}\right)\right\| \leq \frac{\varepsilon}{2\beta}$ by triangle inequality. Therefore, for any unit vector $\mathbf{u}$ and $r \in [0, \frac{1}{2\beta}]$, we have,

$$\left\|\left(\frac{\theta_1^* + \theta_2^*}{2}\right) - \left(\frac{\theta_1 + \theta_2}{2} + r\mathbf{u}\right)\right\| \leq \left\|\left(\frac{\theta_1^* + \theta_2^*}{2}\right) - \left(\frac{\theta_1 + \theta_2}{2}\right)\right\| + \|r\mathbf{u}\| \leq r + \frac{\varepsilon}{2\beta} \leq \frac{1 + \varepsilon}{2\beta} \leq \frac{1}{\beta} .$$

Hence, $\frac{\theta_1 + \theta_2}{2} + r\mathbf{u} \in \mathcal{B}\left(\frac{\theta_1^* + \theta_2^*}{2}, \frac{1}{\beta}\right) \forall$ unit vectors $\mathbf{u}$ and $r \in [0, \frac{1}{\beta}]$. In other words, $\mathcal{B}\left(\frac{\theta_1 + \theta_2}{2}, \frac{1}{2\beta}\right) \subset \mathcal{B}\left(\frac{\theta_1^* + \theta_2^*}{2}, \frac{1}{\beta}\right) \subset \Theta$.

$\qquad\square$

## B.4   Discussion on Exponential Family

Here we discuss some nice properties of the exponential family and also list some examples which satisfy the set of assumptions this work makes.

**Proposition 20.** *Let $\mathcal{E}_\theta(X) := h(X)\exp\left(\theta^\top T(X) - W(\theta)\right)$ be an exponential family distribution. Then we have, (i) $\nabla W(\theta) = \mathbb{E}_{X \sim \mathcal{E}_\theta}[T(X)]$, and (ii) $\nabla^2 W(\theta) = \mathbf{Cov}_{X \sim \mathcal{E}_\theta}[T(X)]$, where $\mathbf{Cov}[x] = \mathbb{E}[(x - \mathbb{E}[x])(x - \mathbb{E}[x])^\top]$.*

*Remark* 6 (Lipschitzness of $W$). From the above lemma, it is clear that Lipschitzness of $W(\theta)$ for $\theta \in \Theta$ is equivalent to assuming that $\|\mathbb{E}_{X \sim \mathcal{E}_\theta}[T(X)]\|$ is bounded for all $\theta \in \Theta$. This is a natural assumption that is satisfied for several distributions under mild assumption on $\Theta$. Moreover, without Lipschitzness it may be difficult to show that if $\theta$ and $\theta'$ are close, then their distributions are close, a property that we expect to hold in a learning problem. Finally, Table 3 enlists some common exponential family distributions and shows what the several assumptions on the exponential family look like.

| Distribution | $\mathcal{E}_\theta(X)$ | $\nabla W(\theta)$ | $\nabla^2 W(\theta)$ | $\Theta$ |
|---|---|---|---|---|
| Exponential | $\exp(\theta X + \log(-\theta))$, $\theta < 0$ | $-\frac{1}{\theta}$ | $\frac{1}{\theta^2}$ | $\kappa \le \frac{1}{\theta^2} \le \lambda \implies \Theta = [-\frac{1}{\sqrt{\kappa}}, -\frac{1}{\sqrt{\lambda}}]$, $L = \sqrt{\lambda} \vee 1$ |
| Weibull (known $k$) | $X^{k-1} \exp(\theta \cdot X^k + \log(k) + \log(-\theta))$, $\theta < 0$ | $-\frac{1}{\theta}$ | $\frac{1}{\theta^2}$ | $\kappa \le \frac{1}{\theta^2} \le \lambda \implies \Theta = [-\frac{1}{\sqrt{\kappa}}, -\frac{1}{\sqrt{\lambda}}]$, $L = \sqrt{\lambda} \vee 1$ |
| Continuous Bernoulli | $\exp\left(\theta X - \log\left(\frac{e^\theta - 1}{\theta}\right)\right)$ | $\frac{e^\theta(\theta-1)+1}{\theta(e^\theta-1)}$ | $\frac{(e^\theta-1)^2 - \theta^2 e^\theta}{\theta^2(e^\theta-1)^2}$ | $|W'(\theta)| \le 1, |W''(\theta)| \le \frac{1}{12} \, \forall \theta \in \mathbb{R}$, $W''(\theta) \ge \kappa \implies \Theta = [-f(\kappa), f(\kappa)]$ for some $f$, $L = 1$ |
| Continuous Poisson | $\frac{\exp(\theta X - W(\theta))}{\Gamma(X+1)}$ | $e^\theta$ | $e^\theta$ | $\kappa \le e^\theta \le \lambda \implies \Theta = [\log(\kappa), \log(\lambda)]$, $L = \lambda \vee 1$ |
| Multivariate Gaussian | $\frac{\exp\left(-\frac{1}{2}(x-\mu)^\mathsf{T}\Sigma^{-1}(x-\mu)\right)}{(\det(2\pi\Sigma))^{1/2}}$, $\theta = \begin{bmatrix} \Sigma^{-1}\mu \\ \Sigma^{-1} \end{bmatrix}$, $T(X) = \begin{bmatrix} X \\ -\frac{1}{2}XX^\mathsf{T} \end{bmatrix}$ | $\begin{bmatrix} \mu \\ -\frac{1}{2}(\Sigma + \mu\mu^\mathsf{T}) \end{bmatrix}$ | Complicated, see Lemma 3 in [DGTZ18] | $\left\|[\mu, -\frac{1}{2}(\Sigma + \mu\mu^\mathsf{T})]^\mathsf{T}\right\| \le L \implies$ $\mu \in \mathcal{B}(0, r_1(L))$ and $\Sigma \in \mathcal{B}(0, r_2(L))$ for some $r(L) > 0$ depending on $L$ |

Table 3: Different Exponential Family Distributions that satisfy the assumption under corresponding $\Theta$ as defined.

# C  Concentration Lemmas

## C.1  Concentration of the Negative Binomial Distribution

**Lemma 21** (Theorem 2.1 [Jan18])**.** *Let $N = \sum_{i=1}^\tau N_i$, where $\tau \ge 1$, and $N_i, \; i \in [\tau]$ are independent and identically distributed Geometric random variables with parameter $p \in (0,1)$, or in other words, $N_i \overset{iid}{\sim} \mathrm{Geom}(p)$. Then, for any $\lambda \ge 1$,*

$$\mathbb{P}\left[N \ge \frac{\lambda\tau}{p}\right] \le e^{-\tau(\lambda - 1 - \log\lambda)}$$

**Lemma 22** (Theorem 3.1 [Jan18])**.** *Using notations from Lemma 21 and for any $\lambda \le 1$,*

$$\mathbb{P}\left[N \le \frac{\lambda\tau}{p}\right] \le e^{-\tau(\lambda - 1 - \log\lambda)}$$

**Lemma 23.** *Using notations from Lemma 21, for any $\varepsilon \in [0,1]$,*

$$\mathbb{P}\left[(1-\varepsilon)\frac{\tau}{p} \le N \le (1+\varepsilon)\frac{\tau}{p}\right] \ge 1 - 2e^{-\frac{\tau}{4}\varepsilon^2}$$

*Proof.*

$$\mathbb{P}\left[(1-\varepsilon)\frac{\tau}{p} \leq N \leq (1+\varepsilon)\frac{\tau}{p}\right] = 1 - \mathbb{P}\left[N \geq (1+\varepsilon)\frac{\tau}{p}\right] - \mathbb{P}\left[N \leq (1-\varepsilon)\frac{\tau}{p}\right]$$

$$\geq 1 - e^{\tau(\ln(1+\varepsilon)-\varepsilon)} - e^{\tau(\varepsilon+\ln(1-\varepsilon))}$$

(setting $\lambda = 1+\varepsilon$ in Lemma 21 and $\lambda = 1-\varepsilon$ in Lemma 22)

$$\geq 1 - 2e^{-\frac{\tau}{4}\varepsilon^2}$$

(via Lemma 27)

$\square$

**Lemma 24.** *Given some $\varepsilon, \delta \in (0,1)$, let Algorithm 6 be run on a $\mathrm{Ber}(p)$ distribution, with $\tau = \frac{4\log(2/\delta)}{\varepsilon^2}$. Then the following holds with probability at least $1 - \delta$:*

$$(1-\varepsilon)\frac{\tau}{p} \leq N \leq (1+\varepsilon)\frac{\tau}{p} . \tag{8}$$

*Proof.* We start by noting that Algorithm 6 keeps sampling from $\mathrm{Ber}(p)$ till $\tau$ 1's are observed. Therefore, the number of iterations needed by the algorithm is $N = \sum_{i=1}^{\tau} N_i$, where $N_i \overset{\text{iid}}{\sim} \mathrm{Geom}(p)$, are iid geometric random variables, with parameter $p$. Essentially, $N_i$ is the number of samples needed between observing the $(i-1)$-th and $i$-th 1.

Applying Lemma 23 finishes the proof. $\square$

**Corollary 25.** *Using notations of Lemma 24, the estimate $\widehat{p} := \frac{\tau}{N}$ returned by Algorithm 6 satisfies $(1-\varepsilon) \leq \frac{\widehat{p}}{p} \leq (1+\varepsilon)$ with probability at least $1 - \delta$.*

*Proof.* This follows from straightforward rearrangement of (8) and the fact that $\frac{1}{1-\varepsilon} \leq 1+\varepsilon$ and $\frac{1}{1+\varepsilon} \geq 1-\varepsilon$ for all $\varepsilon \in [0,1]$. $\square$

## C.2   Supporting Lemmas

**Lemma 26.** *For any $\varepsilon \in [0,1]$, we have the following inequalities: $(1+\varepsilon) \leq e^{\varepsilon - \frac{\varepsilon^2}{4}}$ and $(1-\varepsilon) \leq e^{-\varepsilon - \frac{\varepsilon^2}{4}}$*

*Proof.* Let $f_1(\varepsilon) = \varepsilon - \frac{\varepsilon^2}{4} - \ln(1+\varepsilon)$. Taking derivative of $f_1$, we get

$$f_1'(\varepsilon) = 1 - \left(\frac{\varepsilon}{2} + \frac{1}{1+\varepsilon}\right)$$

$$= 1 - \left(\frac{\varepsilon^2 + \varepsilon + 2}{2 + 2\varepsilon}\right)$$

$$= \frac{\varepsilon - \varepsilon^2}{2 + 2\varepsilon} \geq 0 \qquad (0 \leq \varepsilon \leq 1)$$

Further, $f_1(0) = 0$. Therefore, $f_1(\varepsilon) \geq 0 \ \forall \varepsilon \in [0,1]$. Similarly, one can argue that $f_2(\varepsilon) := -\varepsilon - \frac{\varepsilon^2}{4} - \ln(1-\varepsilon) \geq 0 \ \forall \varepsilon \in [0,1]$. $\square$

**Lemma 27.** *For any non-negative integer $\tau$ and for an $\varepsilon \in [0,1]$, $e^{\tau(\varepsilon+\ln(1-\varepsilon))} + e^{\tau(\ln(1+\varepsilon)-\varepsilon)} \leq 2e^{-\frac{\tau}{4}\varepsilon^2}$*

**Algorithm 6** Sampling from Bernoulli distribution till $\tau$ successes

**Input:** Desired number of successes $\tau$
Set $m = \tau$
**Initialize:** $N = 0$
**while** $m > 0$ **do**
    $Y \sim \text{Ber}(p)$
    $N \leftarrow N + 1$
    **if** $Y = 1$ **then**
        $m \leftarrow m - 1$
**Return:** $\tau/N$

---

*Proof.* Let $f(\varepsilon) := 2e^{-\frac{\tau}{4}\varepsilon^2} - e^{\tau(\varepsilon + \ln(1-\varepsilon))} - e^{\tau(\ln(1+\varepsilon) - \varepsilon)}$. Simplifying,

$$
\begin{aligned}
f(\varepsilon) &= 2e^{-\frac{\tau}{4}\varepsilon^2} - e^{\varepsilon\tau}(1-\varepsilon)^\tau - e^{-\varepsilon\tau}(1+\varepsilon)^\tau \\
&\geq 2e^{-\frac{\tau}{4}\varepsilon^2} - e^{\varepsilon\tau}e^{\tau(-\varepsilon - \frac{\varepsilon^2}{4})} - e^{-\varepsilon\tau}e^{\tau(\varepsilon - \frac{\varepsilon^2}{4})} \qquad \text{(Using Lemma 26)} \\
&\geq 0
\end{aligned}
$$

$\square$

**Lemma 28** (Chernoff Bound). *Let* $X = \sum_{i=1}^{n} X_i$, *where* $X_i \sim \text{Ber}(p_i)$, *and all* $X_i$ *are independent. Let* $\mu = \mathbb{E}[X] = \sum_{i=1}^{n} p_i$. *Then,*

$$
\mathbb{P}\left[|X - \mu| \geq \varepsilon\mu\right] \leq 2\exp\left(-\frac{\mu\varepsilon^2}{3}\right) \qquad \text{for all } 0 < \delta < 1 .
$$

# D Further Experiments

All codes for experiments can be found at https://github.com/nirjhar-das/fairness_audit_partial_feedback.

## D.1 Experiments for the Blackbox Model

### D.1.1 Datasets

We use the Adult Income [BK96] and the Law School [WRC98] datasets to evaluate and compare the performance of our proposed `RS-Audit` (Algorithm 1) with the Baseline (Algorithm 3). On these classification datasets, we use a classifier to act as a decision-maker. The datasets already contain the true labels.

**Adult Income Dataset**

- The Adult Income dataset is a popular classification benchmark dataset for evaluating fairness. The description of features of the dataset is provided in Table 4. The true label $Y$ is a binary variable which determines whether the annual income of a person exceeds $50000 based on various demographic characteristics. For our setting, we choose the 'sex' of the person as the sensitive attribute $A$, and the remaining covariates as features $X$.

- The dataset consists of $48842$ instances, each described via $15$ attributes, of which $6$ are numerical, $7$ are categorical and $2$ are binary attributes.

| Feature Name | Feature Type | Feature Range | Description |
|---|---|---|---|
| age | Numerical | [17 - 90] | Age of an individual |
| workclass | Categorical | 7 | Employment status |
| fnlwgt | Numerical | [13,492 - 1,490,400] | Final weight |
| education | Categorical | 16 | Highest education level |
| educational-num | Numerical | 1 - 16 | Education level (numeric) |
| marital-status | Categorical | 7 | Marital status |
| occupation | Categorical | 14 | Occupation type |
| relationship | Categorical | 6 | Relationship to others |
| race | Categorical | 5 | Race |
| gender | Binary | {Male, Female} | Biological sex |
| capital-gain | Numerical | [0 - 99,999] | Capital gains |
| capital-loss | Numerical | [0 - 4,356] | Capital loss |
| hours-per-week | Numerical | [1 - 99] | Hours worked per week |
| native-country | Categorical | 41 | Country of origin |
| income | Binary | $\{\leq 50K, > 50K\}$ | Income level |

Table 4: Adult Income Dataset description.

| Feature Name | Feature Type | Feature Range | Description |
|---|---|---|---|
| decile1b | Numerical | [1.0 - 10.0] | Student's decile in Year 1 |
| decile3 | Numerical | [1.0 - 10.0] | Student's decile in Year 3 |
| lsat | Numerical | [11.0 - 48.0] | Student's LSAT score |
| ugpa | Numerical | [1.5 - 4.0] | Student's undergraduate GPA |
| zfygpa | Numerical | [-3.35 - 3.48] | First-year law school GPA |
| zgpa | Numerical | [-6.44 - 4.01] | Cumulative law school GPA |
| fulltime | Binary | {1, 2} | Full-time or part-time status |
| fam inc | Categorical | 5 | Family income bracket |
| male | Binary | {0, 1} | Male or female |
| tier | Categorical | 6 | Tier |
| racetxt | Categorical | 6 | Race |
| pass bar | Binary | {0, 1} | Passed bar exam on first try |

Table 5: Law School Dataset description.

**Law School Dataset**

- The Law school dataset (feature description in Table 5) contains the law school admission records across 163 law schools in the United States in the year 1991. The true label $Y$ is a binary variable which determines whether a candidate would pass the bar exam in the first attempt or not. We define the sensitive attribute $A$ to be the sex of the student (male or female). The remaining features act as $X$.

- The dataset contains information of 20798 students characterized by 12 attributes (3 categorical, 3 binary and 6 numerical attributes).

### D.1.2 Experimental Setup

We first train a classifier $f$ on a subsample of the dataset obtained by randomly selecting some samples from the dataset (100 for Adult, 5000 for Law). The classifier is trained to predict $Y$. We consider a wide range of classifiers, namely, logistic regression (LR) based classifiers trained on all the covariates including $A$ (all_LR), all covariates except $A$ (wo_A_LR), and a fair LR classifier trained via Exponentiated Gradient method [ABD$^+$18] (fair). We also consider a random classifier that

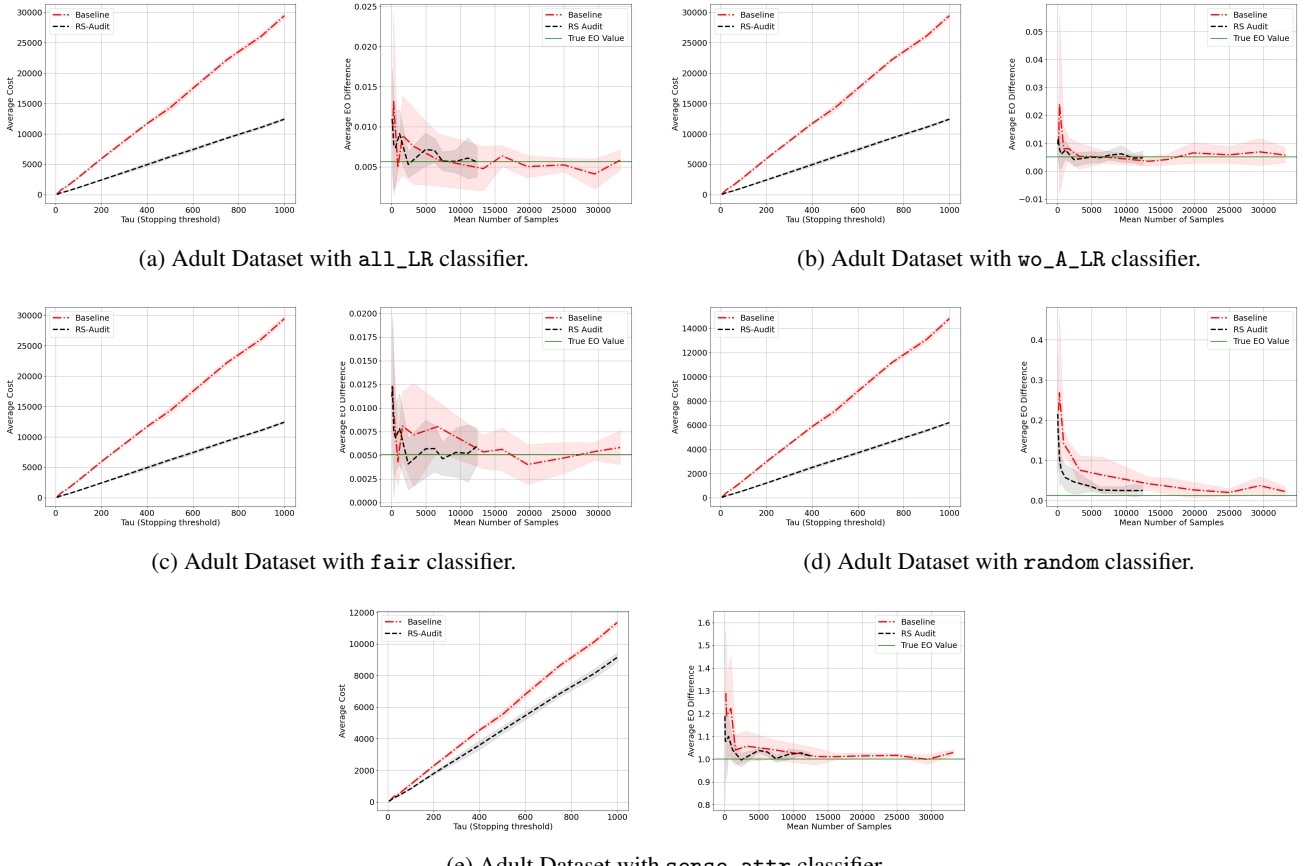

(a) Adult Dataset with `all_LR` classifier.

(b) Adult Dataset with `wo_A_LR` classifier.

(c) Adult Dataset with `fair` classifier.

(d) Adult Dataset with `random` classifier.

(e) Adult Dataset with `sense_attr` classifier.

Figure 2: Results on the Adult Income dataset with different classifiers

assigns outcome based on an independent draw from $\mathrm{Ber}(1/2)$ (`random`) and a classifier that assigns the outcome equal to $A$ (`sense_attr`). Once trained, these classifiers are frozen across various runs of the two algorithms of interest.

Next we take an inference of the classifier on the whole dataset and add the classification as a column to the dataset. To simulate historical data, we use the inference on the whole of the dataset (since past data is available for free). To simulate online samples, we randomly sample a row of the data and feed it to the algorithms. We vary the threshold $\tau$ over the set $\{5, 10, 50, 100, 200, 500, 1000\}$ and repeat the execution of the two algorithms over 5 seeds $\{1092, 42, 13, 729, 333\}$. At the end of execution of every algorithm, we collect the number of labels requested by the algorithm and the cost of the algorithm. We set $c_{lab} = 1$, hence cost corresponds to the number of $Y = 0$ labels requested by the algorithm. We also collect the predicted EOD ($\widehat{\Delta}$) of the algorithm. The true EOD is calculated over the entire dataset.

### D.1.3 Results

The results of the experiments are shown in Figures 2 and 3. We show two plots for each pair of classifier and dataset choice. On the left of each subplot, we show the average cost of the two algorithms `RS-Audit` and Baseline with respect to the stopping threshold $\tau$. We also plot the standard deviation of the cost but it is too small to be significantly visible in the plots. On the right

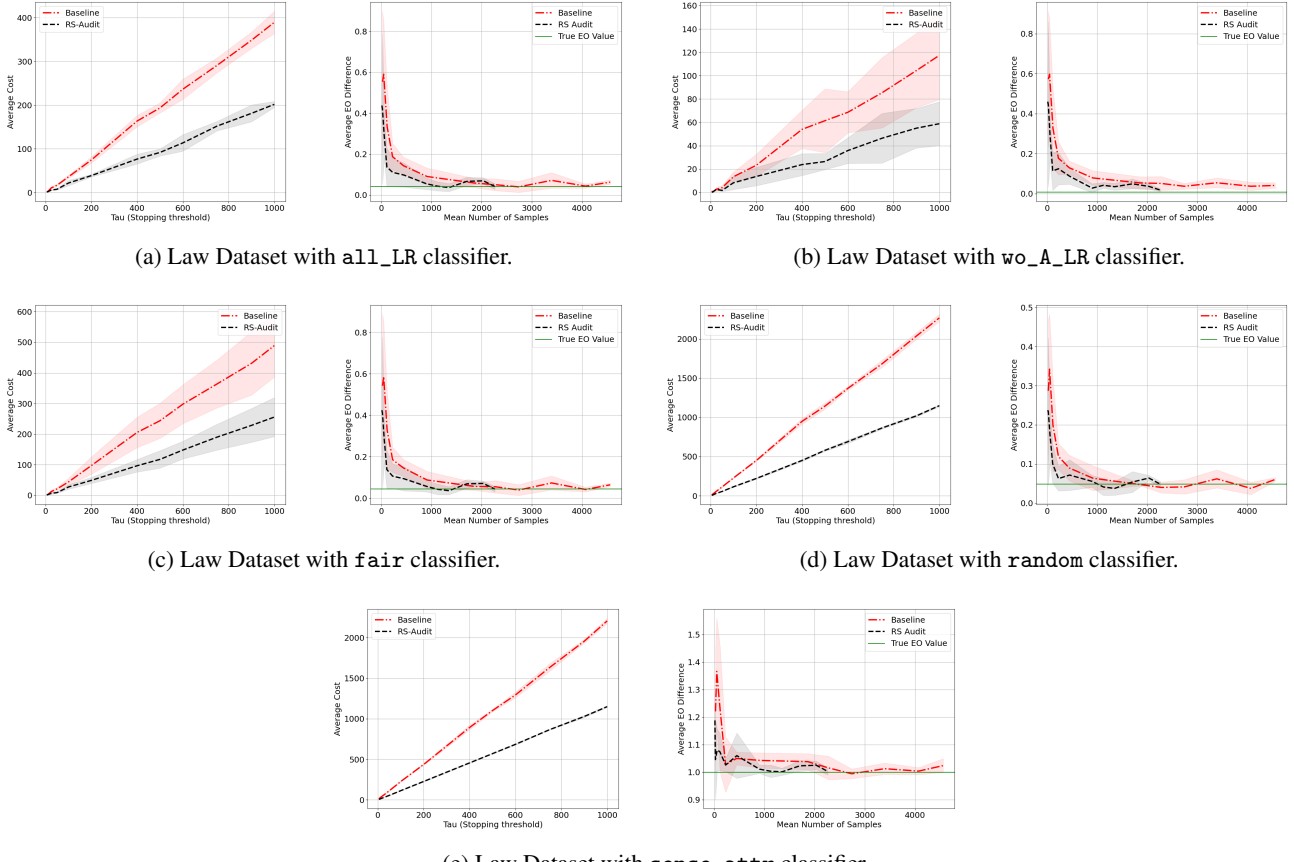

(a) Law Dataset with `all_LR` classifier.

(b) Law Dataset with `wo_A_LR` classifier.

(c) Law Dataset with `fair` classifier.

(d) Law Dataset with `random` classifier.

(e) Law Dataset with `sense_attr` classifier.

Figure 3: Results on the Law School dataset with different classifiers

of each subplot, we show the average $\widehat{\Delta}$ (predicted EOD) versus the average number of requested labels for every $\tau$. The plot also shows the true EOD. From the plots, we observe that the cost of `RS-Audit` is less than of Baseline by about $50\%$ and that `RS-Audit` converges to the true EOD value with a much smaller number of samples, thus providing empirical validation to our theoretical results.

## D.2 Experiments for the Mixture Model

### D.2.1 Instance Construction

For the experiments under the mixture model, we construct a synthetic instance that follows the modeling assumptions (see Section 2 and 3). In our experiments, we fix $\mathcal{A} = \{0, 1\}$. We set $\mathbb{P}[A = 1] = 0.7$ and $\mathbb{P}[Y = 1 \mid A = 1] = 0.4$. We choose spherical Gaussian in $\mathbb{R}^d$, $d = 5$, as our exponential family, with covariance $\sigma^2 = 4$ across all subgroups $(y, a) \in \{0, 1\} \times \mathcal{A}$. The means are generated as follows. For a particular $a \in \mathcal{A}$, we first choose $\mu_{1,a} \in \mathbb{R}^d$ with each coordinate of $\mu_{1,a}$ from uniform distribution over $[-1, 1]$. Thereafter, we sample unit vector $\mathbf{u}$ uniformly randomly and set $\mu_{0,a} \coloneqq \mu_{1,a} + r \cdot \mathbf{u}$, where $r = \Omega\left(\sqrt{\log\left(\frac{1}{\varepsilon}\right)}\right)$ as specified in the separation condition (1).

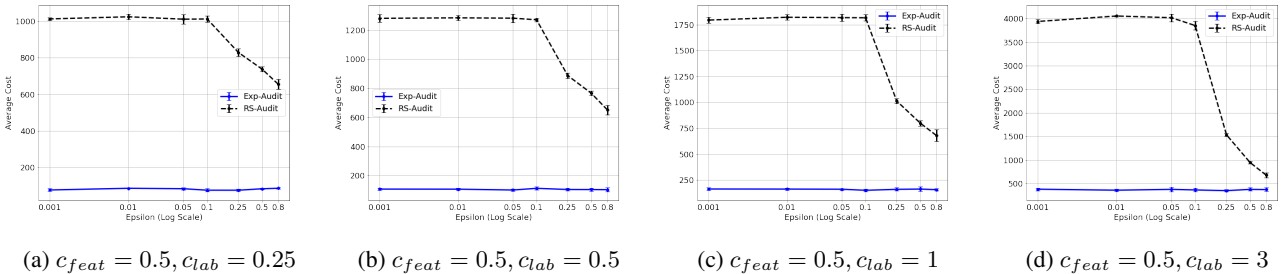

(a) $c_{feat} = 0.5, c_{lab} = 0.25$      (b) $c_{feat} = 0.5, c_{lab} = 0.5$      (c) $c_{feat} = 0.5, c_{lab} = 1$      (d) $c_{feat} = 0.5, c_{lab} = 3$

Figure 4: Results on the Mixture Model for different choices of $c_{feat}$ and $c_{lab}$.

We vary $\varepsilon$ over the range of values $\{0.8, 0.5, 0.25, 0.1, 0.05, 0.01, 0.001\}$ and $(c_{feat}, c_{lab})$ over the set $\{(0.5, 0.25), (0.5, 0.5), (0.5, 1), (0.5, 3)\}$. For every combination of $\varepsilon$ and $(c_{feat}, c_{lab})$, we repeat the runs of the algorithm 5 times. We compare the algorithms RS-Audit (Algorithm 1) and Exp-Audit (Algorithm 2). To be able to complete the experiment withing reasonable time, we capped $\tau$ for RS-Audit at 1000 and $R$ in Exp-Audit at 20000. To simulate past data, we first sample 200000 points as follows: first sample $A \sim \text{Ber}(0.7)$, then sample $Y \sim \text{Ber}(q_{1|A})$ and then $X \sim \mathcal{E}_{\theta^*_{Y,A}}$. Here, $q_{1|1} = 0.7$ and $q_{1|0} = 0.4$. Then, we take an inference of the classifier on these points. The positively classified points were sent to the TruncEst subroutine (Line 4, Algorithm 2). This past data generation process is also repeated with every repetition of the algorithm.

### D.2.2 Results

In Fig. 4, we plot the cost of the two algorithms averaged over the different runs against $\varepsilon$. In every subplot, the plot shows the average cost of the two algorithms. We also show the standard deviation of each observation with error bars. We observe that the cost of Exp-Audit is nearly independent of $\varepsilon$ whereas the cost of RS-Audit increases with decrease in $\varepsilon$. We also measure the fraction of the times the algorithms are able to predict the correct hypothesis in Table 6. From the table, we observe that Exp-Audit is correct at almost all times whereas RS-Audit has some error in the high $\varepsilon = 0.8$ case. Note that in the case that $\varepsilon = 0.25$, the assumption of the hypothesis test is not satisfied, *i.e.*, the true EOD is neither 0 nor $> \varepsilon$. Moreover, in other cases, where the algorithms are mostly correct, the true EOD is either $< \frac{\varepsilon}{2}$ or $> \varepsilon$. This is supported by our theory since we show that both the algorithms satisfy $|\widehat{\Delta} - \Delta| \leq \frac{\varepsilon}{2}$ (see Eq. (5)).

| Epsilon ($\varepsilon$) | True Hypothesis | Exp-Audit **Correctness** | RS-Audit **Correctness** |
|:---:|:---:|:---:|:---:|
| 0.8 | FAIR | $1.0 \pm 0.0$ | $0.8 \pm 0.4$ |
| 0.5 | FAIR | $1.0 \pm 0.0$ | $1.0 \pm 0.0$ |
| 0.25 | FAIR | $0.0 \pm 0.0$ | $0.2 \pm 0.4$ |
| 0.1 | UNFAIR | $1.0 \pm 0.0$ | $1.0 \pm 0.0$ |
| 0.05 | UNFAIR | $1.0 \pm 0.0$ | $1.0 \pm 0.0$ |
| 0.01 | UNFAIR | $1.0 \pm 0.0$ | $1.0 \pm 0.0$ |
| 0.001 | UNFAIR | $1.0 \pm 0.0$ | $1.0 \pm 0.0$ |

Table 6: Audit correctness of Exp-Audit and RS-Audit across different values of $\epsilon$ and true hypotheses.

