# OpenReview forum: "Cost Efficient Fairness Audit Under Partial Feedback"
_NeurIPS.cc/2025/Workshop/Reliable_ML — NeurIPS 2025 - Reliable ML Workshop_

### Official Review · Reviewer_vKxx · 2025-09-17
**The authors provide an interesting model for fairness under partial feedback, with strong theoretical foundations across different regimes and compelling experimental results.**

**Rating:** 7
**Confidence:** 3

**Review:**

This work considers a new fairness model for classification under partial feedback that takes into account the cost of acquiring labels for instances that would be rejected by the classifier. The authors provide algorithms and upper and lower bounds for both the setting with no distributional assumptions and for mixtures of exponential families.

Strengths:

1.Novel proposed model that is more realistic for real-world applications.
2.Proposed algorithms for both distributional and agnostic settings with tight lower and upper bounds.
3.Promising experimental results.

Weaknesses/Suggestions:

1.In real-world scenarios, such as bank loan applications, it would be more realistic if the cost function for acquiring a label varied across different contexts.
2.Another term for partial feedback is "apple tasting feedback." It would be interesting to consider as a baseline an Apple Tasting learning algorithm (e.g., Deterministic Apple Tasting by Zachary Chase and Idan Mehalel).
3.In some real-world applications, the dual problem may be more applicable: given a fixed budget, what are the best fairness guarantees achievable?
4.An online version of this problem could also be of interest.

---

### Official Review · Reviewer_Hewr · 2025-09-20
**Review for paper : Cost Efficient Fairness Audit Under Partial Feedback**

**Rating:** 8
**Confidence:** 4

**Review:**

The authors study fairness auditing under partial information feedback, considering two distributional settings: (i) black-box models and (ii) exponential families. They assume that only positively classified data are initially available, while negatively classified examples can be obtained at an additional cost. To address this challenge, the paper introduces a model that formalizes the auditing cost, distinguishing between two types of expenses for unobserved data: acquiring a feature vector and acquiring its corresponding label.
As a baseline, the authors first present a natural algorithm. In the black-box model, they establish a lower bound on the number of requested samples required to achieve epsilon-delta fairness, and propose an algorithm that matches this bound up to logarithmic factors. In the exponential families setting, they develop an algorithm with a provable upper bound on the auditing cost. Finally, they demonstrate their results through simulations.

**Strengths**
The paper clearly explains notation, definitions, and theorems, making the technical content accessible. The authors also provide motivating examples that connect the definitions to real-world applications.
- A novel cost model for fairness auditing is introduced and well-grounded through concrete examples.
- The work provides strong theoretical guarantees, including both lower and upper bounds in different settings.

**Weaknesses**
The paper does not exhibit significant weaknesses. However, potential directions for future work include:
- Providing a stronger justification for the chosen baseline algorithm, or considering alternative baselines.
- Establishing a lower bound in the exponential families setting to complement the provided upper bound.

---

### Official Review · Reviewer_Eomh · 2025-09-23
**Review of Cost Efficient Fairness Audit Under Partial Feedback**

**Rating:** 7
**Confidence:** 3

**Review:**

**Summary**

The paper studies the problem of conducting fairness audits by requesting labeled samples at some cost. They introduce a cost model that captures differing costs for collecting features vs labels with the cost for labels being much greater.The goal is to provide an audit that is approximately correct with high probability, with a low cost of the samples requested. They provide an algorithm that has the best worst case number of samples collected to solve the fairness audit problem. They also consider problems with more structure, namely a mixture model satisfying some assumptions satisfied by common distribution families. For problems with this structure, they develop algorithms that make use fo techniques for learning with truncated sampling and show that the audit cost can be lower than the worst case audit cost. They compare their algorithms with baseline algorithms such as requesting random samples on some benchmarks.

**Strengths**

1) The connection made between the problem and learning with truncated samples is interesting.

2) The paper main ideas behind the proofs are explained intuitively.

**Weaknesses**

1) The matching upper and lower bounds from Theorems 2 and 3 are shown in terms of number of samples rather than costs. However, improvement for structured families compared to worst-case is in terms of costs. This inconsistency makes it hard to see the logical claims the paper makes.

2) The fairness problem the paper solves distinguishes no unfairness from greater than \epsilon unfairness. A more natural problem is distinguishing between less than \epsilon unfairness and perhaps greater than 2\epsilon unfairness. It is unclear if the results rely too much on the null that unfairness is exactly zero.

**Suggestions**

1) Rewriting the upper and lower bounds in Theorems 2 and 3 in terms of costs rather than number of samples